# Development and Evaluation of EDTA-Treated Rabbits for Bioavailability Study of Chelating Drugs Using Levofloxacin, Ciprofloxacin, Hemiacetal Ester Prodrugs, and Tetracycline

**DOI:** 10.3390/pharmaceutics15061589

**Published:** 2023-05-24

**Authors:** Yorinobu Maeda, Honoka Teraoka, Ami Okada, Mirei Yamamoto, Shintaro Natsuyama, Yuhzo Hieda, Yuka Nagatsuka, Yuhki Sato, Takeshi Goromaru, Teruo Murakami

**Affiliations:** 1Laboratory of Drug Information Analytics, Faculty of Pharmacy & Pharmaceutical Sciences, Fukuyama University, Hiroshima 729-0292, Japan; 2Common Resources Center, Faculty of Pharmacy & Pharmaceutical Sciences, Fukuyama University, Hiroshima 729-0292, Japan; 3Laboratory of Clinical Evaluation of Drug Efficacy, Faculty of Pharmacy & Pharmaceutical Sciences, Fukuyama University, Hiroshima 729-0292, Japan; 4Faculty of Pharmaceutical Sciences, Hiroshima International University, Hiroshima 739-2631, Japan

**Keywords:** rabbits, coprophagy, EDTA-treated rabbits, gastric metals, chelate formation, adsorption, levofloxacin, ciprofloxacin, hemiacetal ester prodrug, tetracycline

## Abstract

Laboratory rabbits are fed foods rich with cationic metals, and while fasting cannot empty gastric contents because of their coprophagic habits. This implies that, in rabbits, the oral bioavailability of chelating drugs could be modulated by the slow gastric emptying rates and the interaction (chelation, adsorption) with gastric metals. In the present study, we tried to develop a rabbit model with low amounts of cationic metals in the stomach for preclinical oral bioavailability studies of chelating drugs. The elimination of gastric metals was achieved by preventing food intake and coprophagy and administering a low concentration of EDTA 2Na solution one day before experiments. Control rabbits were fasted but coprophagy was not prevented. The efficacy of rabbits treated with EDTA 2Na was evaluated by comparing the gastric contents, gastric metal contents and gastric pH between EDTA-treated and control rabbits. The treatment with more than 10 mL of 1 mg/mL EDTA 2Na solution decreased the amounts of gastric contents, cationic metals and gastric pH, without causing mucosal damage. The absolute oral bioavailabilities (mean values) of levofloxacin (LFX), ciprofloxacin (CFX) and tetracycline hydrochloride (TC), chelating antibiotics, were significantly higher in EDTA-treated rabbits than those in control rabbits as follows: 119.0 vs. 87.2%, 9.37 vs. 13.7%, and 4.90 vs. 2.59%, respectively. The oral bioavailabilities of these drugs were significantly decreased when Al(OH)_3_ was administered concomitantly in both control and EDTA-treated rabbits. In contrast, the absolute oral bioavailabilities of ethoxycarbonyl 1-ethyl hemiacetal ester (EHE) prodrugs of LFX and CFX (LFX-EHE, CFX-EHE), which are non-chelating prodrugs at least in in vitro condition, were comparable between control and EDTA-treated rabbits irrespective of the presence of Al(OH)_3_, although some variation was observed among rabbits. The oral bioavailabilities of LFX and CFX from their EHE prodrugs were comparable with LFX and CFX alone, respectively, even in the presence of Al(OH)_3_. In conclusion, LFX, CFX and TC exhibited higher oral bioavailabilities in EDTA-treated rabbits than in control rabbits, indicating that the oral bioavailabilities of these chelating drugs are reduced in untreated rabbits. In conclusion, EDTA-treated rabbits were found to exhibit low gastric contents including metals and low gastric pH, without causing mucosal damage. Ester prodrug of CFX was effective in preventing chelate formation with Al(OH)_3_ in vitro and in vivo, as well as in the case of ester prodrugs of LFX. EDTA-treated rabbits are expected to provide great advantages in preclinical oral bioavailability studies of various drugs and dosage formulations. However, a marked interspecies difference was still observed in the oral bioavailability of CFX and TC between EDTA-treated rabbits and humans, possibly due to the contribution of adsorptive interaction in rabbits. Further study is necessary to seek out the usefulness of the EDTA-treated rabbit with less gastric contents and metals as an experimental animal.

## 1. Introduction

Rabbits have significant advantages as experimental animals in preclinical pharmacokinetic studies, as compared with small animals such as mice, rats and guinea pigs. For example, rabbits are low-cost, have a rapid growth rate, are easy to manage, and repeated blood sampling can be made in rabbits without anesthesia for a crossover study. In addition, most of the dosage formulations for human use can be examined in rabbits. In contrast, however, rabbits perform coprophagy directly from the anus, and one or two days’ fasting cannot empty gastric contents. The stomach of rabbits is always full of contents such as feces, food, hair, and so on, in contrast to the case of rats and humans [1,2,3]. Coprophagy is necessary to supply many essential nutrients and maintain gut flora for rabbits, and the prevention of coprophagy significantly reduces the growth rate of rabbits and alters cholesterol absorption and bile acid composition [2,3,4,5]. Accordingly, the use of rabbits for preclinical oral bioavailability studies is limited, because of the slow gastric emptying rate (GER) of orally administered drugs. The slow GER of drugs in rabbits can modify the metabolic rates and therefore bioavailability, especially of lipophilic compounds, as compared with fasted dogs and humans [6,7]. To overcome the slow GER in rabbits, rabbits having empty stomachs, or stomach (or gastric)-emptying controlled rabbits, were developed by the muzzle method or the cangue method. Various findings regarding GER including the effect of GER on oral bioavailability have been reported [6,8,9,10,11,12,13].

In general, laboratory rabbits are fed foods rich in cationic metals in addition to various essential nutrients, suggesting that the gastrointestinal contents of rabbits contain various cationic metals at higher concentrations due to coprophagy and fullness. Thus, it is speculated that the oral bioavailability of chelating drugs could be reduced by the interaction with various cationic metals in the gastrointestinal lumen, in addition to the delayed GER in rabbits. Various antibiotics such as fluoroquinolones, macrolides, polypeptide antibacterial agents, sulfonamides, and tetracyclines are known to form chelate complexes with metal cations in water and physicochemical adsorption to metal cation-containing solid materials [14,15,16]. In addition, the oral bioavailability of these chelating drugs could be varied among rabbits and foods due to the difference in cationic metal composition in foods among different manufacturing companies.

In the present study, we tried to develop a rabbit model with a low amount of cationic metals in the stomach for preclinical oral bioavailability studies of chelating drugs. Elimination of gastric metals was achieved by administering a low concentration of disodium ethylenediamine tetraacetic acid (EDTA 2Na) orally. EDTA 2Na, a typical chelating agent, can chelate various polyvalent metal cations and form water-soluble chelate complexes in water. The low toxicity of EDTA 2Na is also recognized. For example, in EDTA chelation therapy, 3 g of EDTA 2Na or EDTA-Ca is infused intravenously to eliminate toxic metals such as lead (Pb), cadmium (Cd), gadolinium (Gd), mercury (Hg), and so on in the body into urine [17,18]. In the treatment of rabbits with EDTA 2Na orally, or EDTA-treated rabbits, rabbits were fasted with free access to 1 mg/mL EDTA 2Na solution and coprophagy was prevented for one day before experiments. Control rabbits were fasted but coprophagy was not prevented. The usefulness of EDTA-treated rabbits as preclinical experimental animals was evaluated by comparing the gastric contents, gastric metal contents and gastric pH between EDTA-treated and control rabbits. In addition, oral bioavailabilities of chelating drugs were compared between EDTA-treated and control rabbits. As chelating drugs, antibiotics such as levofloxacin (LFX), ciprofloxacin (CFX) and tetracycline (TC) were used, because these antibiotics are known to interact with cationic metals and cationic metal-containing solid materials by chelation and adsorption [19,20,21,22,23,24,25,26,27,28,29]. Recently, it was reported that among patients who received oral fluoroquinolones or tetracyclines, 47 patients (26.6%) ingested antibiotics with divalent or trivalent cation-containing compounds, in which ciprofloxacin (61.7%) was the most used antibiotic, followed by moxifloxacin (25.5%) and doxycycline (12.8%) [30]. TC, a potent chelating and adsorptive antibiotic, is known to interact with various cationic metal-containing biomaterials and cause side effects such as pigmented teeth and skeletal pigmentation by physiochemical adsorption, in addition to chelate formation in water [20,21,22,23,24,25,26,27]. Ethoxycarbonyl 1-ethyl hemiacetal ester (EHE) prodrugs of LFX (LFX-EHE) and CFX (CFX-EHE) were also used since ester prodrugs of LFX such as LFX-EHE are known to avoid chelate formation in vitro and in vivo [31,32,33]. CFX-EHE was newly synthesized in the present study. The contribution of rabbit’s gastric metals on the oral bioavailability of these antibiotics including LFX-EHE and CFX-EHE was evaluated by comparing the oral bioavailability in the absence and presence of aluminum hydroxide (Al(OH)_3_), a widely used antacid, in control and EDTA-treated rabbits. The establishment of EDTA-treated rabbits with less gastric contents including metal contents is expected to provide great advantages in preclinical oral bioavailability studies of various drugs, and formulations. The chemical structures of LFX, LFX-EHE, CFX, CFX-EHE, and TC used in the present study are shown in Figure 1.

## 2. Materials and Methods

### 2.1. Materials

The following materials (manufacturing company or reagent company, location) were used: LFX (Apollo Scientific, Cheshire, UK), CFX (LKT Laboratories, Inc., Saint Paul, MN, USA), TC hydrochloride (Fujifilm Wako Pure Chemical Corporation, Osaka, Japan), 1-chloroethyl ethyl carbonate (Tokyo Chemical Industry Co., Ltd., Tokyo, Japan), dried Al(OH)_3_ gel (Pfizer, Tokyo, Japan), anhydrous aluminum chloride (AlCl_3_, Wako Pure Chemical Ind, Ltd., Osaka, Japan), Mueller–Hinton broth (Nippon Becton Dickinson Co., Ltd., Tokyo, Japan), deuterated chloroform (CDCl_3_, Kanto Chemical, Co. Inc., Tokyo, Japan), hexadeuterodimethyl sulfoxide (dimethyl sulfoxide-d_6_, DMSO-d_6,_ Kanto Chemical, Co. Inc. Tokyo, Japan), filter paper (Advantec^®^, Tokyo Roshi Kaisha, Ltd., Tokyo, Japan), and syringe filter with 0.22 μm of pore size (Millipore, Tokyo, Japan). Bacteria such as *Staphylococcus* (*S.*) *aureus* NBRC 15035 (ATCC 29213) and *Escherichia* (*E.*) *coli* NBRC 15034 (ATCC 25922) were provided by Biological Resource Center Department of Biotechnology, National Institute of Technology and Evaluation (Chiba, Japan)*. Pseudomonas* (*P.*) *aeruginosa* JCM 6119 (ATCC 27853) were provided by the RIKEN BRC through the National BioResource Project of the MEXT (Tokyo, Japan). The high-pressure liquid chromatography (HPLC) column (L-column 2 ODS, 150 mm × 4.6 mm, i.d., 5 μm) was purchased from CERI Co., Ltd. (Saitama, Japan). Other materials including organic solvents for HPLC used were of the highest grade available. 

### 2.2. Synthesis of LFX-EHE

LFX-EHE was synthesized in the same manner as reported previously according to the reported modified method [33,34]. Briefly, 1-chloroethyl ethyl carbonate (8 mmol) was added to N, N-dimethylformamide (DMF) solution containing anhydrous potassium carbonate (4 mmol) and LFX (2 mmol). The mixture solution was stirred for 2 h under argon gas bubbling. The product was extracted with ethyl acetate, washed with distilled water, dehydrated with anhydrous sodium sulfate, and evaporated to dryness. Recrystallization of crude LFX-EHE was made from diethyl ether. Crystal precipitated was filtrated with filter paper (Advantec^®^, Tokyo, Japan) and dried at 40 °C for 5 h. Melting point (MP) was measured on the Yanagimoto micro-melting point apparatus (MP-S3, J-Science Lab. Co., Ltd., Kyoto, Japan) and was uncorrected. Proton nuclear magnetic resonance (^1^H-NMR) spectra were recorded on a JEOL JNM-400S at 400 MHz (JEOL Ltd., Tokyo, Japan), and chemical shifts relative to tetramethylsilane (TMS, an internal standard, δ 0.00) were estimated. NMR spectra were measured in CDCl_3_ (δ 7.26 ppm). The carbon nuclear magnetic resonance (^13^C-NMR) spectroscopic data were recorded with a JEOL JNM-400S at 101 MHz (JEOL Ltd., Tokyo, Japan), and chemical shifts relative to CDCl_3_ (δ = 77.0 ppm) and DMSO-d_6_ (δ 39.7) were estimated. The mass spectra of the prodrugs were recorded on a JEOL JMS-700 spectrometer (JEOL Ltd., Tokyo, Japan) through the direct inlet system.

### 2.3. Synthesis of CFX-EHE

CFX-EHE was newly synthesized in the present study via Boc-CFX and Boc-CFX-EHE, according to the reported modified method for ampicillin prodrugs [34]. The purity of the synthesized compounds was examined by thin layer chromatography (TLC, Kieselge1 60 F254 plates, Sigma-Aldrich Japan, Tokyo, Japan) using ethanol-ammonia water (3:1, *v*/*v*) as a developing solvent. Detection of CFX, Boc-CFX, Boc-CFX-EHE, and CFX-EHE was made by short-wave ultraviolet light at 254 nm. The Rf values of CFX, Boc-CFX, Boc-CFX-EHE, and CFX-EHE were 0.08, 0.35, 0.78, and 0.24, respectively. The synthetic process of CFX-EHE is summarized in Figure 2.

#### 2.3.1. Synthesis of Boc-CFX

Boc-CFX was synthesized by the reported method [35]. Briefly, CFX (500 mg/1.51 mmol) was dissolved in 1 mol/L NaOH solution (5 mL) and tetrahydrofuran (THF, 10 mL) was added, followed by the dropwise addition of di-tert-butyl dicarbonate (Boc_2_O, 360 mg/1.66 mmol) in THF (10 mL) and stirred at room temperature for 16 h. The solvent was removed under reduced pressure and the resulting material was diluted in H_2_O and neutralized with a saturated NH_4_Cl solution. The precipitate was collected by vacuum filtration and washed with H_2_O to afford the product as a white amorphous solid (606 mg, 93%). ^1^H NMR (CDCl_3_) δ: 1.19–1.23 (2H, m), 1.37–1.43 (2H, m), 1.50 (9H, s), 3.28–3.30 (4H, m), 3.51–3.56 (1H, m), 3.66–3.68 (4H, m), 7.36 (1H, d, J = 7.3 Hz), 8.03 (1H, d, J = 12.8 Hz), 8.77 (1H, s). ^13^C-NMR (CDCl_3_) δ: 8.2, 28.4, 35.3, 49.7, 80.4, 105.0, 108.1, 112.5, 112.8, 120.3, 120.3, 139.0, 145.7, 145.8, 147.6, 152.4, 154.6, 154.9, 166.8, 177.1. MS *m*/*z*: 431(M+). Chemical formula: C_22_H_26_FN_3_O_5_. Molecular weight: 431.4644.

#### 2.3.2. Synthesis of Boc-CFX-EHE

Boc-CFX-EHE was synthesized by the reported modified method [34]. Anhydrous potassium carbonate (553 mg/4 mmol) was added to the Boc-CFX (862 mg/2 mmol) solution dissolved in DMF (25 mL) with vigorous stirring in the round bottom flask. Then, 1-chloroethyl ethyl carbonate (8 mmol) was added and stirred for 2 h at 70 °C under an argon atmosphere. Crushed ice and H_2_O (120 mL) were slowly added to the mixture after 2 h-stirring and cooling. Crystal precipitated was filtrated with filter paper and was dried at 40 °C for 5 h. The synthesized yield of Boc-CFX-EHE was 82% (632 mg). MP. 196–198 °C (dec.). ^1^H-NMR (CDCl_3_) δ: 1.10–1.14 (2H, m), 1.29–1.35 (5H, m), 1.49 (9H, s), 1.66 (3H, d, J = 5.0 Hz), 3.21 (4H, t, J = 5.0 Hz), 3.40–3.43 (1H, m), 3.65 (4H, t, J = 5.0 Hz), 4.22 (2H, qd, J = 1.3, 7.1 Hz), 7.00 (1H, q, J = 5.3 Hz), 7.26 (1H, q, J = 4.8 Hz), 8.03 (1H, d, J = 12.8 Hz), 8.51 (1H, s). ^13^C-NMR (CDCl_3_) δ: 8.2, 14.1, 19.8, 28.4, 34.6, 49.9, 64.4, 80.2, 91.6, 105.0, 109.1, 113.4, 113.6, 123.2, 123.3, 137.9, 144.5, 144.6, 148.6, 152.2, 153.1, 154.6, 154.7, 163.1, 172.9. MS *m*/*z*: 547(M+). Chemical formula: C_27_H_34_FN_3_O_8_. Molecular weight: 547.5804.

#### 2.3.3. Synthesis of CFX-EHE

Boc-CFX-EHE (500 mg/0.91 mmol) was dissolved in anhydrous dichloromethane (DCM, 15 mL) and cooled to 0 °C. Anhydrous anisole (100 μL) was added followed by the dropwise addition of trifluoroacetic acid (TFA, 4 mL). The reaction mixture was stirred at 0 °C for 30 min, warmed to room temperature, and stirred for a further 40 min. The solvent was removed under a stream of nitrogen gas (N_2_) and the resulting gum was triturated with ice-cold ethyl acetate. The precipitate was collected, diluted in H_2_O and DCM, and basified to pH 9 with a 3% NaHCO_3_ solution. The product was extracted with chloroform, washed with distilled water, dehydrated with anhydrous sodium sulfate, and the organic solvent was evaporated to dryness. Recrystallization of CFX-EHE was made from ethyl acetate. The synthesized yield of CFX-EHE was 56% (291 mg). MP. > 300 °C (dec.). ^1^H-NMR (DMSO-d_6_) δ: 1.05–1.15 (2H, m), 1.21 (3H, t, J = 7.1 Hz), 1.25–1.27 (2H, m), 1.53 (3H, d, J = 5.5 Hz), 3.25 (4H, br s), 3.39–3.40 (4H, m), 3.65–3.71 (1H, m), 4.15 (2H, q, J = 7.0 Hz), 6.78 (1H, q, J = 5.5 Hz), 7.46 (1H, d, J = 7.3 Hz), 7.77 (1H, d, J = 13.3 Hz), 8.44 (1H, s). ^13^C-NMR (CDCl_3_) δ: 8.1, 8.1, 14.4, 19.9, 35.6, 43.5, 47.6, 64.6, 91.7, 107.3, 108.0, 112.2, 112.4, 116.2, 119.1, 122.8, 122.8, 138.5, 143.6, 143.7, 149.6, 151.9, 153.1, 154.3, 158.7, 159.0, 162.7, 172.3. MS *m*/*z*: 447(M+). Chemical formula: C_22_H_26_FN_3_O_6_. Molecular weight: 447.4634. Spectrum of ^1^H-NMR (A), and ^13^C-NMR (B) of CFX; spectrum of ^1^H-NMR (A), and Mass (B) of Boc-CFX; spectrum of ^1^H-NMR (A), ^13^C-NMR (B) and Mass (C) of Boc-CFX-EHE; and spectrum of ^1^H-NMR (A), ^13^C-NMR (B) and Mass (C) of CFX-EHE are shown in Appendix A.

### 2.4. Partition Coefficients

Partition coefficients of CFX and CFX-EHE were determined in the same manner as reported previously [32,33]. Briefly, chloroform and 0.1 M Tris-HCl buffer (pH 6.5) were mutually saturated before experiments. CFX and CFX-EHE were dissolved in 0.1 M Tris-HCl buffer (100 μg/mL each). Four milliliters of each drug solution were added to 4 mL of chloroform, and the mixture was vigorously shaken for 30 min at 25 °C. After centrifugation at 3000 rpm for 10 min, the concentration of the compound in both phases was determined by HPLC.

### 2.5. Solubility

Solubility of CFX and CFX-EHE was determined at pH 2 buffer adjusted with hydrochloric acid and 0.1 M Tris-HCl buffer (pH 6.5) in the same manner as reported previously [32,33]. An excess amount of CFX and CFX-EHE (approximately 100 mg) was suspended and stirred vigorously for 30 min at 25 °C. The mixture was centrifuged at 3000 rpm for 10 min, and the supernatant was filtered through a syringe filter with 0.22 μm of pore size (Millipore, Tokyo, Japan), and the concentrations of the compound in the filtrate were measured by HPLC.

### 2.6. Chelation with Al^3+^ Ion In Vitro

Chelating potencies of CFX, CFX-EHE and TC were determined in the same manner as reported previously [32,33]. Briefly, CFX, CFX-EHE, and TC were dissolved in 0.5 mM Tris-HCl buffer (pH 6.5) at a concentration of 100 μM, respectively. AlCl_3_ was also dissolved in the same buffer at a concentration of 50 μM, 100 μM, 200 μM, 1 mM, or 2 mM, and each drug solution and AlCl_3_ solution were mixed at an equal volume. The mixture solution was shaken, stood for 30 min at 24 °C and centrifuged. The supernatant was filtrated through a syringe filter with 0.22 μm of pore size (Millipore, Tokyo, Japan). The concentrations of CFX in each filtrate were determined by HPLC after hydrolyzing CFX-EHE with 1 mol/L NaOH containing 1.0 mg/mL EDTA-2Na.

### 2.7. Chemical and Enzymatic Stability

The chemical and enzymatic stability of CFX-EHE, an ester prodrug of CFX, was evaluated in the same manner as reported previously [32,33]. Briefly, to examine the chemical stability of CFX-EHE, pH 6.5 and pH 7.4 phosphate buffer solutions (PBS) were used. Enzymatic stability of CFX-EHE (the initial concentration was 0.1 mM) was determined at 1, 5, and 15 min after the start of incubation at 37 °C using the following specimen: rat 10% plasma obtained from male Sprague Dawley (SD) rats, rat 2% small intestinal mucosal homogenates, and rat 2% liver homogenates in the same manner as described previously [32,33]. Male SD rats were purchased from Japan SLC, Inc. (Hamamatsu, Japan), and kept in stainless steel cages equipped with an automatic water supply and excrement flushing device. Each biological specimen was prepared using pH 7.4, 25 mM Tris-HCl buffer. Concentrations of CFX produced from CFX-EHE ester prodrug in the reaction mixture were determined periodically after the start of incubation, in which the further metabolic reaction after sampling was stopped by adding methanol at a volume ratio of 1:2.

### 2.8. Determination of Minimum Inhibitory Concentrations (MICs) of CFX and CFX-EHE

MIC values of CFX and CFX-EHE were determined by agar plate dilution method using Mueller–Hinton broth in the same manner as reported previously [32,33]. Briefly, CFX (10 mg) and CFX-EHE (10 mg) were dissolved in 1 mL of 0.1 M hydrochloric acid, respectively, and 4 mL of 0. l M pH 7.4 phosphate buffer was added to each solution. The final concentrations of CFX and CFX-EHE in the incubation medium were adjusted to 62.50, 31.25, 15.63, 7.81, 3.91, 1.95, 0.98, 0.49, 0.24, 0.12, 0.06, 0.03, 0.016, 0.008, 0.004 0.002, and 0.001 μg/mL, respectively. The bacterial numbers of *S. aureus*, *E. coli*, and *P. aeruginosa* were adjusted to McFarland 0.5 (1.5 × 10^6^ cells/mL) with the sterilized isotonic NaCl solution. These bacterial solutions were inoculated in Mueller–Hinton broth containing CFX or CFX-EHE. Facultative anaerobic and aerobic bacteria were cultured under the aerobic condition at 37 °C for 20 h.

### 2.9. Development of EDTA-Treated Rabbits

Male albino rabbits weighing 2.5 ± 0.5 kg were purchased from Japan SLC, Inc. (Hamamatsu, Japan), and kept in stainless steel cages equipped with an automatic water supply and excrement flushing device. The room was maintained in a 12:12-h dark-light cycle and kept at a constant temperature of 22 ± 0.5 °C. Rabbits were supplied solid food for rabbits (CR-3M, CLEA Japan, Inc., Tokyo, Japan), and the average amount of CR-3M that each rabbit ingested was 100–110 g/day. CR-3M contains various vitamins and cationic metals including Ca, Mg, and iron (Fe^2+^, Fe^3+^) in addition to various energy sources. One day before experiments, control rabbits were fasted with free access to water by keeping them in the stainless-steel cages as described above. Coprophagy was not prevented in control rabbits. In the case of EDTA-treated rabbits, rabbits were moved to wooden rabbit fixing boxes (KN-319-A, Natsume Seisakusho Co., Ltd., Tokyo, Japan) one day before experiments, in which the neck is fixed to prevent food intake and coprophagy but with free access to 1 mg/mL EDTA 2Na aqueous solution. In addition, 1 h before the bioavailability study of antibiotics, control rabbits received 10 mL of water orally and EDTA-treated rabbits received 10 mL of water containing 1 mg/mL EDTA 2Na by stomach intubation, respectively.

#### 2.9.1. Estimation of Gastric Contents, Gastric Ca and Mg Concentrations, and Gastric pH

To estimate gastric contents, gastric Ca and Mg concentrations, and gastric pH in the untreated control (not fasted), control (fasted) and EDTA-treated rabbits, rabbits were sacrificed by intraperitoneal injection of an excess amount of pentobarbital. Rabbits were opened along the midline and their stomach was isolated. The stomach contents were taken out into a test tube and weighed, and the suspension was centrifuged at 3000 rpm for 10 min to measure the amount of gastric water. Then, concentrations of the Ca and Mg in the supernatant were measured by using L type Wako Ca (Fujifilm, Tokyo, Japan) and L type Wako Mg N (Fujifilm, Tokyo, Japan), respectively, and an automated clinical chemistry analyzer, TBA-nx360 (Canon, Tokyo, Japan). The pH of gastric water was also determined.

#### 2.9.2. Evaluation of Mucosal Damage after EDTA Treatment

To evaluate the plausible mucosal damage or change in membrane permeability after treatment with EDTA 2Na, the oral absorption of fluorescein isothiocyanate (FITC) dextran with an average molecular weight of 4400 (FD-4), a poorly absorbable marker compound due to the hydrophilic macromolecule [36], was compared between control and EDTA-treated rabbits. FD-4 was administered orally by stomach intubation at 10 mg/kg and 10 mL water. Blood (100 μL each) was taken periodically at 0, 0.5, 1, 2, and 4 h after administration from the ear vein. Blood samples were centrifuged at 15,000× *g* for 5 min at 4 °C. The concentrations of FD-4 in plasma were determined by Spectro fluorophotometer RF-1500 (Shimadzu Co., Ltd., Kyoto, Japan) using an excitation and emission wavelength of 480 nm and 520 nm, respectively [37].

#### 2.9.3. Oral Bioavailability of Fluoroquinolones in Control and EDTA-Treated Rabbits

Either LFX 20 mg/kg (0.055 mmol/kg), LFX-EHE 26.4 mg/kg (0.055 mmol/kg), CFX 30 mg/kg (0.091 mmol/kg), or CFX-EHE 40.5 mg/kg (0.091 mmol/kg) was administered orally by stomach intubation without (control) or with Al(OH)_3_ (100 mg/kg) in control and EDTA-treated rabbits as follows; administration of Al(OH)_3_ suspension (approximately 3 mL) and water (approximately 2 mL) to wash the tube line, and then drug solution (approximately 3 mL) followed by water (approximately 2 mL). The total amount of water ingested was 10 mL. After administration of each drug, blood (100 μL each) was taken periodically from ear veins. Blood samples were centrifuged at 3000× *g* for 10 min, and plasma samples were taken and kept in a deep freezer until analysis.

#### 2.9.4. Oral Bioavailability of TC in Control and EDTA-Treated Rabbits

TC hydrochloride was administered orally at a dose of 150 mg/kg together with 10 mL water to control and EDTA-treated rabbits. In a separate experiment, 1 mg/mL EDTA 2Na aqueous solution was administered 1 day before administration of each drug at a volume of 40 mL (20 mL × 2, in the morning and evening) for treatment, in addition to the free access to the drinking water containing 1 mg/mL EDTA 2Na. After administration orally by stomach intubation, blood samples (100 μL each) were taken periodically from the ear vein at 0.25, 0.5, 1, 2, 4 and 6 h. Blood samples were centrifuged at 3000× *g* for 10 min to obtain plasma samples.

#### 2.9.5. Intravenous Administration of LFX, CFX and TC in Control Rabbits

To estimate the absolute oral bioavailability (F) of LFX (20 mg/kg), CFX (30 mg/kg) and TC (10 mg/kg), respectively, antibiotics were administered intravenously by a constant-rate infusion for 2 min 50 s into the ear vein. Blood was taken periodically at 3 min, 15 min, 30 min, 1 h, 2 h, 4 h and 6 h after the start of constant-rate infusion of each antibiotic from the opposite ear vein. Dosing solutions of LFX, CFX and TC were prepared by dissolving in saline at a concentration of 4 mg/kg/mL, 6 mg/kg/mL and 2 mg/kg/mL, respectively. The area under the concentration-time curve (AUC) from 0 to infinity (AUC_0–∞_) values of each drug in plasma after intravenous (AUC_0–∞, iv_) and oral (AUC_0–∞, po_) administrations were estimated as follows: AUC_0–∞_ = AUC_0–xh_ + C(xh)/β, where AUC_0–xh_ (or AUC from 0 to x h, in which x h is the final time of blood sampling) was estimated by trapezoidal rule and AUC after blood sampling (AUC_xh–∞_) was estimated by dividing the blood concentration at x h, C(xh), with the slope (β) of the terminal phase. The calculation of AUC was made by using Excel. The plasma concentrations of LFX were determined for 6 h after intravenous and oral administrations, those of CFX were determined for 6 h after intravenous and for 4 h after oral administration, respectively, and those of TC were determined for 4 h after intravenous and for 6 h after oral administration, respectively. The absolute oral bioavailability (F) was estimated by using the following equation: F = AUC_0–∞,po_ × Dose_iv_/(AUC_0–∞,iv_ × Dose_po_), where Dose_po_ and Dose_iv_ represent the dose of each drug for oral administration and intravenous infusion, respectively.

### 2.10. Analysis of LFX, CFX and TC by HPLC

Concentrations of LFX in plasma samples were determined by HPLC using CFX as an internal standard (IS) in the same manner as reported previously [32,33]. In the case of plasma samples (each 100 μL) containing LFX-EH, 0.05 mL of 1.0 mol/L NaOH was added and incubated for 30 min and then neutralized with 0.05 mL of 1.0 mol/L HCl solution before analysis with HPLC. The solution was filtered with a syringe-type filter to obtain a filtrate. The column used was an L-column 2 ODS (150 mm × 4.6 mm, i.d., 5 μm, CERI Co., Ltd., Saitama, Japan). The mobile phase was a mixture of acetonitrile and 0.3% triethylamine adjusted at pH 3.3 with phosphoric acid (16:84 *v*/*v*). The flow rate of the mobile phase was 1 mL/min. The detection was made at UV 295 nm.

Concentrations of CFX in aqueous solution and plasma samples were determined by HPLC using LFX as IS according to the reported modified method [38]. Briefly, each CFX-containing sample (50 μL) was mixed with 100 μL of methanol containing 5.0 μg/mL LFX. The mixture was mixed by a vortex mixer for 30 s and centrifuged at 15,000× *g* for 5 min at 4 °C, and a 20 μL aliquot of the supernatant was injected into the HPLC column (L-column 2 ODS, 150 mm × 4.6 mm, CERI Co., Ltd., Saitama, Japan). In the case of CFX-EHE aqueous samples (each 1.0 mL) obtained from chelation, partition coefficients and solubility studies, 0.5 mL of 1.0 mol/L NaOH was added, incubated for 30 min and then neutralized with 0.5 mL of 1.0 mol/L HCl solution. The solution was filtered with a syringe-type filter to obtain a filtrate. The mobile phase of HPLC was a mixture of acetonitrile and water containing 0.3% triethylamine (pH 3.3 adjusted with phosphoric acid) (16:84 *v*/*v*). The flow rate of the mobile phase was 1 mL/min. Detection was made by a fluorescence detector at 278 nm and 450 nm for an excitation and emission wavelength, respectively [39].

The concentrations of TC in aqueous solution and plasma were determined by HPLC using LFX as IS according to the modified method [40]. Briefly, each TC-containing sample (50 μL) was mixed with 100 μL of methanol containing 5.0 μg/mL LFX. The mixture was vortex-mixed for 30 s and centrifuged at 15,000× *g* for 5 min at 4 °C, and a 20 μL aliquot of the supernatant was injected into the HPLC column (L-column 2 ODS (150 mm × 4.6 mm, i.d., 5 μm, CERI Co., Ltd., Saitama, Japan). The mobile phase was a mixture of acetonitrile and 0.3% triethylamine adjusted at pH 3.3 with phosphoric acid (16:84 *v*/*v*). The flow rate of the mobile phase was 1 mL/min. The detection was made at 273 nm.

The HPLC system (Shimadzu, Kyoto, Japan) consisted of a model LC-20AD pump, a fixed injection loop of 20 μL, and a fluorescence spectrophotometer (Shimadzu, RF-10A_XL_). Data acquisition was performed with the Sept 3000’s processor (Hangzhou, China). Examples of chromatograms of LFX, CFX, and TC in rabbits’ plasm are shown in Figure 3.

### 2.11. Statistical Analysis

The data were expressed as the mean ± standard deviation (SD). Statistical analysis was performed using the unpaired Student’s *t*-test. *p*-values less than 0.05 were regarded as statistically significant.

## 3. Results

### 3.1. Synthesis of CFX-EHE

CFX-EHE (Figure 1d) was newly synthesized in the present study via Boc-CFX and Boc-CFX-EHE, according to the reported modified method [34]. The synthesized yield of CFX-EHE was 56%, and the MP was >300 °C (dec.). The purity of CFX-EHE was high since the ^l^H-NMR spectral data displayed only specific signals of CFX-EHE. The purity of CFX-EHE was also evaluated by TLC using a mixture of ethanol-ammonia water (3:1, *v*/*v*) as a developing solvent. The spot detected using shortwave ultraviolet light (254 nm) was one spot, and the Rf values of CFX and CFX-EHF were 0.08 and 0.24, respectively. Data: ^1^H-NMR (DMSO-d6) δ: 1.05–1.15 (2H, m), 1.21 (3H, t, J = 7.1 Hz), 1.25–1.27 (2H, m), 1.53 (3H, d, J = 5.5 Hz), 3.25 (4H, br s), 3.39–3.40 (4H, m), 3.65–3.71 (1H, m), 4.15 (2H, q, J = 7.0 Hz), 6.78 (1H, q, J = 5.5 Hz), 7.46 (1H, d, J = 7.3 Hz), 7.77 (1H, d, J = 13.3 Hz), 8.44 (1H, s). ^13^C-NMR (CDCl_3_) δ: 8.1, 8.1, 14.4, 19.9, 35.6, 43.5, 47.6, 64.6, 91.7, 107.3, 108.0, 112.2, 112.4, 116.2, 119.1, 122.8, 122.8, 138.5, 143.6, 143.7, 149.6, 151.9, 153.1, 154.3, 158.7, 159.0, 162.7, 172.3. MS *m*/*z*: 447(M+). Chemical formula: C_22_H_26_FN_3_O_6_. Molecular weight: 447.4634.

### 3.2. Physicochemical Properties of CFX-EHE: Comparison with CFX

#### 3.2.1. Partition Coefficient and Solubility

To evaluate the lipophilicity of CFX and CFX-EHE, the apparent partition coefficients (PCs) of CFX and CFX-EHE were determined in a partition system of chloroform and pH 6.5 buffer. Estimated PCs and calculated log *p* values are listed in Table 1. The apparent PC of CFX-EHE was 13-fold higher than CFX. The PC of LFX-EHX (75.2) was also 13-fold higher than LFX (5.6). The solubility of CFX and CFX-EHE were significantly lower than LFX and LFX-EHE, respectively, as follows: at pH 2.0, the solubility of LFX and LFX-EHE was 42.95 mg/mL and 16.54 mg/mL, respectively. [33].

#### 3.2.2. Chelation of CFX, CFX-EHE and TC with Al^3+^ In Vitro

To evaluate the chelating ability of CFX, CFX-EHE and TC, drug (100 μM) solution of each drug was mixed with AlCl_3_ solution (50 μM, 100 μM, 200 μM, 1 mM and 2 mM, respectively). The precipitation percentages of TC were greater than those of CFX. CFX-EHE showed no precipitation for 30 min (Table 2). Previously, we also determined the chelating ability of LFX and LFX-EHE in the same in vitro conditions, in which the precipitation percentage of LFX was 12.4% at 50 μM of AlCl_3_. When compared at a concentration of 50 μM AlCl_3_, the chelating ability of each antibiotic was in the following order: TC > CFX > LFX. In contrast, CFX-EHE was not precipitated even in the presence of 1 mM of AlCl_3_ (drug: AlCl_3_ = 1:20), as well as in the case of LFX-EHE [33]. When EDTA 2Na (4 mg/mL, 12 mM) was added to these mixture solutions of CFX and AlCl_3_, the precipitation was completely inhibited.

#### 3.2.3. Chemical and Enzymatic Stability of CFX-EHE

The stability of orally administered CFX-EHE in the small intestine, intestinal membrane, plasma and liver was evaluated in vitro. CFX-EHE was stable chemically in pH 6.5 and pH 7.4 phosphate-buffered saline (PBS), but rapidly hydrolyzed enzymatically in 10% plasma, 2% intestinal mucosa homogenates and 2% liver homogenates (Table 3). The average hydrolysis % of LFX-EHE in 2% mucosal homogenate, 2% liver homogenates and 10% plasma after 1 min incubation was 17.4%, 17.7% and 8.2%, respectively [33]. CFX-EHE appeared to be more sensitive to enzymatic hydrolysis than LFX-EHE, indicating that CFX-EHE can produce active parent drug, CFX, rapidly when CFX-EHE was absorbed into the intestinal membrane.

#### 3.2.4. The MIC Values of CFX-EHE

The MIC values of CFX-EHE against *S. aureus*, *E. coli*, and *P. aeruginosa* were determined. In the prodrug concept, prodrugs should have lower pharmacological action as compared with their parent drug [41]. The MIC values of CFX-EHE against *S. aureus*, *E. coli*, and *P. aeruginosa* were more than 10 times higher than that of CFX (Table 4). These data indicated that CFX-EHE themselves are pharmacologically not active or have low activity levels as antibiotics. Similarly, the MIC values of LFX-EHE against the above bacteria were also significantly higher than those of LFX [33]. The low pharmacological activities of ester prodrugs would be in good agreement with the prodrug concept [41].

### 3.3. Development of EDTA-Treated Rabbits

Rabbits perform coprophagy directly from the anus [1,2,3]. Eating soft feces can improve feed efficiency and maintain gut flora in rabbits because this coprophagy allows the absorption of nutrients and bacterial fermentation products (amino acids, volatile fatty acids and vitamins B and K), and the digestion of previously undigested food pellets. Thus, the rabbit’s stomach is never empty, with food, cecal pellets and ingested hair present in a loose latticework [5,42]. In general, laboratory rabbits are fed with rich cationic metals. Due to the coprophagic habits of rabbits, a part of these cationic metals could be retained in the stomach contents. To examine the contribution of gastric metals on the oral bioavailability of chelating drugs, in the present study rabbits were pretreated with EDTA 2Na, a representative chelating agent. The concentration of EDTA 2Na was fixed at 1 mg/mL, and the effect of EDTA treatment was evaluated from the viewpoints of Ca and Mg concentrations in stomach contents, and mucosal damage, or intestinal absorption of FD-4, a poorly absorbable hydrophilic compound with an average molecular weight of 4400.

#### 3.3.1. Evaluation of Gastric Contents and Gastric Ca and Mg Concentrations

The amount of gastric contents of rabbits was compared among different treatments. Group 1, untreated control rabbits, had free access to food pellets and drinking water, and coprophagy was not prevented. Group 2, control rabbits, fasted with free access to drinking water one day before the experiment. Coprophagy was not prevented. Ten mL of water was administered 1 h before the experiment. Group 3, EDTA-treated rabbits, fasted with free access to 1 mg/mL EDTA 2Na solution one day before the experiment. Coprophagy was prevented. Ten mL of 1 mg/mL EDTA 2Na solution was also administered 1 h before the experiments. When these Group 1, 2 and 3 rabbits were sacrificed, gastric contents, water contents, gastric pH, and concentrations of Ca and Mg in gastric water were determined (Table 5). The experimental procedures of the sampling of gastric contents and water in control and EDTA-treated rabbits are shown in Figure 4. The amounts of gastric contents and concentrations of Ca and Mg in the gastric water were in the following order: Group 1 > Group 2 > Group 3. The amounts of gastric water and gastric pH were in the following order: Group 1 > Group 2 = Group 3. These results would indicate that fasting and water injection, or EDTA treatment, can facilitate the emptying of gastric contents including gastric metals (Table 5). The total amounts of Ca (a product of stomach water volume and Ca concentrations in the water) of Nos. 1, 2 (untreated control rabbits), Nos. 3, 4 (control rabbits) and Nos. 5–7 (EDTA-treated rabbits) were 79.75 mg, 35.42 mg (average, 57.59 mg), 1.84 mg, 6.75 mg (average, 4.30 mg), and 0.17 mg, 0.06 mg, and 0.19 mg (average, 0.14 mg), respectively.

#### 3.3.2. Oral Bioavailability of FD-4 in Control and EDTA-Treated Rabbits

To evaluate the plausible mucosal damage after treatment with 1 mg/mL EDTA, the plasma concentrations of FD-4 administered orally were compared between control and EDTA-treated rabbits. There was no difference in the plasma concentrations of FD-4 between them, indicating that the membrane permeability to FD-4 was not altered in EDTA-treated rabbits (Table 6).

### 3.4. Oral Bioavailability of LFX, LFX-EHE CFX and CFX-EHE in the Absence and Presence of Al(OH)_3_ in Control and EDTA-Treated Rabbits

#### 3.4.1. Intravenous Administration of LFX, CFX and TC

To estimate the absolute oral bioavailability (F) of LFX, CFX, and TC in rabbits, the AUC_0−∞_ value after intravenous constant-rate infusion for 2.83 min of each antibiotic was estimated. The doses of LFX and CFX for intravenous administration were the same as those for oral administration (20 mg/kg and 30 mg/kg, respectively). On one hand, the intravenous dose of TC was reduced to 10 mg/kg, although the oral dose was 150 mg/kg, based on the results of the preliminary study. Plasma disposition-time courses of LFX, CFX and TC are shown in Figure 4. These drugs were eliminated in a fashion of a two-compartment model. Some pharmacokinetic parameters such as Cmax in plasma (concentration at 3 min after the start of the constant rate infusion), elimination rate constant in the terminal phase (β), AUC_0–6h_ for LFX and CFX or AUC_0–4h_ for TC, and AUC_0−∞_ are listed in Table 7. When the value of Cmax/Dose_iv_ was compared among three antibiotics, the order was as follows; LFX = CFX < TC, indicating that the tissue distribution volume of TC is smaller than those of LFX and CFX, and the distribution volume of LFX and CFX are comparable (Figure 5). Similarly, the total plasma clearance (CL_total_) estimated by Dose_iv_/AUC_0–∞_ were in the following order: CFX (1273 mL/h/kg) = TC (1130 mL/h/kg) > LFX (823 mL/h/kg).

#### 3.4.2. Oral Bioavailability of LFX and LFX-EHE

LFX and LFX-EHE were administered at a dose of 20 mg/kg as LFX, and the dose of Al(OH)_3_ was 100 mg/kg in both control and EDTA-treated rabbits. The time courses of LFX concentrations in plasma after oral administration of LFX and LFX-EHE in the absence and presence of Al(OH)_3_ are shown in Figure 6. Some pharmacokinetic parameters such as Cmax, time to reach Cmax (Tmax), AUC_0–6_, AUC_0–∞_ of plasma LFX, and oral bioavailability (F) of LFX are listed in Table 8. The oral bioavailability of LFX from LFX-EHE was comparable with the oral bioavailability of LFX itself in both control and EDTA-treated rabbits, exhibiting higher oral bioavailabilities. In the absence of Al(OH)_3_, both the rate of oral bioavailability (Cmax and Tmax) and the extent of oral bioavailability (AUC) of LFX in EDTA-treated rabbits were higher than in control rabbits. This indicated that both the rate and extent of oral bioavailability of LFX in control rabbits are significantly lower in control rabbits than in EDTA-treated rabbits. When Al(OH)_3_ was administered concomitantly with LFX, the Cmax, and AUC_0–6_ of LFX were significantly lower than those in the absence of Al(OH)_3_ in both control and EDTA-treated rabbits. In contrast, the rate and extent of oral bioavailability of LFX from LFX-EHE were comparable between control and EDTA-treated rabbits and co-administration of Al(OH)_3_ exerted no significant effects on the rate and extent of oral bioavailability of LFX from LFX-EH. This would indicate that LFX-EHE can prevent chelate formation in vitro and in vivo. Although there was no significant difference in the oral bioavailability between the absence and presence of Al(OH)_3_, however, the average Cmax and AUC of LFX from LFX-EHE in the presence of Al(OH)_3_ appeared to be lower as compared with the results of LFX without Al(OH)_3_ in EDTA-treated rabbits (Table 8).

#### 3.4.3. Oral Bioavailability of CFX and CFX-EHE

The doses of CFX and CFX-EHE were 30 mg/kg as CFX, and the dose of Al(OH)_3_ was 100 mg/kg in both control and EDTA-treated rabbits. Both the rate and extent of oral bioavailability of CFX in EDTA-treated rabbits were significantly higher than in control rabbits, as well as in the case of LFX. These data will indicate that the rate and extent of oral bioavailability of CFX in control rabbits are suppressed by gastric cationic metals, by forming a poorly soluble chelate complex. By co-administrating Al(OH)_3_ with CFX to control and EDTA-treated rabbits, the rate and extent of oral bioavailability of CFX were suppressed. In contrast, the rate and extent of oral bioavailability of CFX-EHE were comparable between control and EDTA-treated rabbits. When the oral bioavailability of CFX from CFX-EHE was compared with the oral bioavailability of CFX itself, there was no significant difference between them in both control and EDTA-treated rabbits, as well as the cases of LFX-EHE. The co-administration of Al(OH)_3_ exerted no significant effects on the rate and extent of oral bioavailability of CFX from CFX-EHE, indicating that CFX-EHE can prevent chelate formation in vivo. However, it would be considered that there may be some interaction such as adsorption between CFX-EHE and Al(OH)_3_, especially in EDTA-treated rabbits (Figure 7 and Table 9).

#### 3.4.4. Oral Bioavailability of TC in Control and EDTA-Treated Rabbits

The dose of TC hydrochloride was fixed at 150 mg/kg in both control and EDTA-treated rabbits. In EDTA-treated rabbits, values of Cmax and AUC_0–6h_ of TC were significantly greater than those in control rabbits, in which the value of Tmax was comparable between both rabbits. In a separate experiment, 20 mL (10 mL on one day before the experiment and 10 mL on the experiment day) of 1 mg/mL EDTA Na was administered orally for treatment, in addition to the free access to the drinking water containing 1 mg/mL EDTA 2Na. The dose of TC hydrochloride was the same (150 mg/kg). In this case, the value of standard deviation (SD) of plasma TC concentrations became smaller than 10 mL single ingestion of 1 mg/mL EDTA 2Na, although there was no significant difference in the average plasma concentrations of TC between 10 mL and 20 mL of EDTA 2Na. This may suggest that the dose of 20 mL of 1 mg/mL EDTA 2Na ingestion is preferred to eliminate the cationic metals from the stomach completely. (Figure 8 and Table 10).

## 4. Discussion

In the present study, we tried to develop EDTA-treated rabbits with a small amount of gastric metals by using a low concentration of EDTA 2Na and the evaluation of their usefulness in oral bioavailability studies of chelating drugs was made by comparing the oral bioavailability between control and EDTA-treated rabbits. The full stomach contents in rabbits are due to the coprophagic habits of rabbits. This consideration implied that the stomach contents of rabbits contain high concentrations of metal cations because rabbits’ food contains high concentrations of various metal cations. In the present study, EDTA-treated rabbits were prepared by keeping rabbits in fixing boxes (in which the neck of rabbits is fixed) to prevent coprophagy, fasting for one day with free access to 1 mg/mL EDTA 2Na, and ingesting 10 mL EDTA 2Na 1 h before the experiment. The EDTA treatment decreased the gastric Ca contents to <1% of untreated control rabbits (Table 5), and the oral bioavailabilities of chelating drugs such as LFX, CFX and TC were significantly higher in EDTA-treated rabbits as compared in control rabbits (Figure 6, Figure 7 and Figure 8). This means that the oral bioavailabilities of chelating drugs in rabbits are suppressed due to the interaction (chelation, adsorption) with gastric metals, in contrast to the case of fasted rats and humans. The present EDTA treatment method was thought to be simple, easy, non-toxic, and low-stress for rabbits. Thus, it was expected that developed EDTA-treated rabbits with less gastric contents, less gastric metal cation contents, and low gastric pH, are useful preclinical experimental animal models as well as fasted rats in studying the oral bioavailability of chelating drugs. However, further studies will be needed to clarify the mechanism of the effect of EDTA treatment on gastric emptying, because the amount of EDTA 2Na used for treatment (>10 mL of 1 mg/mL EDTA 2Na) would be far lower compared to the amounts of gastric metal contents in untreated control rabbits (Table 5). It may be speculated that EDTA forms chelation with cationic metals in gastric water, cleaves the metal-mediated binding of gastric contents and makes porridge-like materials, and facilitates gastric emptying of the slurry materials. In this section, some discussion was made regarding the preparation method of EDTA-treated rabbits, the interaction mechanism of LFX, CFX and TC with cationic metals, and a comparison of the oral bioavailabilities of LFX, CFX and TC between rabbits and humans.

### 4.1. Preparation of EDTA-Treated Rabbits

Laboratory rabbits are fed foods rich in cationic metals in addition to various essential nutrients. In the present study, CR-3M (CLEA Japan, Inc., Fujinomiya, Japan) was used as rabbits’ food and this food contains the following cationic metals: Ca, 2.10–2.13 g; Mg, 0.23–0.26 g; Fe, 38.1–39.1 mg; Mn, 7.90–9.12 mg; Zn, 4.72–5.08 mg; Cu, 0.78–0.96 mg; and so on/100 g food (information from CLEA Japan Inc.). Similarly, another commercially available food for rabbits also contains cationic metals as follows: Ca, 1.2 g; Mg, 0.32 g; Fe, 51.9 mg; Mn, 7.7 mg; Zn, 5 mg; Cu, 1.05 mg; and so on/100 g food (LRC4, Oriental Yeast Co., Ltd., Tokyo, Japan). Rabbits ate 100–110 g of food per day. It was reported that 47 % of the total excreted amount of Ca was excreted in the urine and 53% was in the feces in female New Zealand White rabbits [43]. The stomach of rabbits is always full of contents with feces, unabsorbed food from coprophagy, hair, and so on, and 1–3 days of fasting cannot empty rabbits’ stomachs [1,2,3,5,42]. Taken together, it was suspected that rabbits with coprophagic habits exhibit high concentrations of cationic metals in gastric contents because of the ingestion of food rich in cationic metals. The high concentrations of gastric metals, in addition to full gastric contents, would greatly modulate the oral bioavailability of chelating drugs. More than 40 years ago, rabbits with empty stomachs, or stomach-emptying controlled rabbits, were developed to eliminate gastric contents [8,9]. However, the preparation method of rabbits with an empty stomach was not easy, because feeding for several days with special foods that easily degraded after ingestion, gastric lavage under anesthesia, and keeping rabbits in a cangue to prevent coprophagy for three days were needed before experiments. In the present study, rabbits were required to undergo one-day fasting, ingestion of >10 mL of 1 mg/mL EDTA 2Na solution and one-day prevention of coprophagy to prepare rabbits with less gastric contents and metals and low gastric pH (Table 5).

In developing EDTA-treated rabbits, the concentration of EDTA 2Na was fixed at 1 mg/mL (2.97 mM, 0.1 %) in water, since higher concentrations of EDTA 2Na are known to modify the membrane permeability of the paracellular route by opening the cellular tight junctions between enterocytes. For example, in Caco-2 cells, the coadministration of 2.5 mM EDTA 2Na (0.84 mg/mL, 0.084%) increased the permeability of fluorescein isothiocyanate (FITC) and FITC-dextrans, and produced changes in ZO-1, claudin-1, occludin, and E-cadherin distribution, indicating a tight junction effect. The copresence of 0.25% EDTA 2Na increased the permeability of PEG 4000 to 14 times the control [44,45]. In contrast, however, in in vivo conditions, a higher concentration of EDTA 2Na is needed to increase the paracellular transport to hydrophilic compounds. It was reported that 5 mg/mL (0.5%) EDTA 2Na increased the intestinal permeation of enalaprilat (349 Da) and hexarelin (887 Da), but did not increase them at 1 mg/mL (0.1%) EDTA 2Na in rats [46]. Other researchers reported that the minimal effective dose of EDTA 2Na was 10 mg/kg to increase the membrane permeability to 4-amino-1-hydroxybutylidene-1,1-bisphosphonate, a bisphosphonate, and the lowest effective dose was 100 mg/kg EDTA 2Na to increase the permeability of dichloromethylene bisphosphonate in rats [47]. Similarly, EDTA 2Na is known to increase the paracellular transport of hydrophilic compounds in a concentration-dependent manner in Caco-2 cells in vitro and in rats in situ and in vivo. It was also reported that, after the application of high EDTA concentrations of more than 6 mM (2.02 mg/mL) for 45 min, the transepithelial electrical resistance (TEER) did not recover to the TEER value before EDTA treatment, but, at the lower concentrations of EDTA such as 2 and 3 mM (1.01 mg/mL), the TEER recovered again while the permeability of sodium fluorescein remained at an elevated level in Caco-2 cells [46,48,49,50]. In in vivo conditions, the decreased TEER recovers to the TEER value before EDTA treatment due to the re-distribution of cationic metals from blood circulation, and the elevated permeability was rapidly decreased to the level in the absence of EDTA 2Na. In addition, the chelated EDTA 2Na does not modify TEER any more as follows: the presence of 1% (10 mg/mL) EDTA 2Na increased the rectal absorption of sodium ampicillin but 5% (50 mg/mL) EDTA-Ca did not increase the rectal absorption of sodium ampicillin in rabbits [51]. In the case of rabbits, gastric water contains various cationic metals at high concentrations. EDTA 2Na ingested will be easily inactivated by forming chelation with gastric metals in the stomach, which may explain the lower cytotoxicity of 1 mg/mL EDTA treatment in rabbits (Table 6).

As shown in Table 5, the amounts of gastric contents and water in control rabbits (No. 3 and 4 rabbits) were lower as compared to untreated control rabbits (No. 1 and 2 rabbits), although the amounts in control rabbits were scattered between two rabbits. The scattering of the amounts of gastric contents in control rabbits may suggest that the one-day fasting of rabbits and the administration of 10 mL of water may be effective in facilitating the emptying of the gastric contents more or less as compared with untreated control rabbits. It was also thought that not all fasted rabbits eat coprophagy every day. Further study is necessary regarding the effect of fasting and water ingestion, but it was found that treatment of rabbits with >10 mL of 1 mg/mL EDTA 2Na orally can decrease the gastric total contents including cationic metals and gastric water without causing mucosal damage (Table 5 and Table 6). The usefulness of EDTA-treated rabbits in pharmacokinetic studies of chelating drugs was evaluated by comparing the oral bioavailabilities of LFX, CFX, TC, LFX-EHE and CFX-EHE in the absence and presence of Al(OH)_3_ in control and EDTA-treated rabbits. LFX, CFX and TC are known to form chelate complexes with various cationic metals. In addition, CFX and TC are also known to adsorb various metal-containing solid materials [37,52,53,54,55]. The physicochemical adsorptive potencies of TC and CFT are reported to be as follows: TC > CFX, because TC contains more functional groups for bonding than CFX [53]. In contrast, the adsorptive potency of LFX, as well as ofloxacin (OFX), to solid Al(OH)_3_ was reported to be weak among quinolones [56].

### 4.2. Regarding LFX and LFX-EHE

In Biopharmaceutics Classification System (BCS), LFX was classified as a BCS Class 1 drug with high solubility and high permeability [57]. The oral bioavailabilities of LFX reported in humans are as follows: approximately 100% in a review article [58], or 69 ± 7% at a dose of 200 mg [59], approximately 70% and fecal recovery of LFX accounted for approximately 15% of an intravenous dose [60], 82 ± 13% in patients with AIDS at a dose of 500 mg [61], and 79 ± 47% at a dose of 250 mg in healthy volunteers [62]. In rats, approximately 90% of LFX dose was absorbed as the intact form from the intestinal tract into the portal system, and in sheep, the absolute oral bioavailability of LFX was 114 ± 27.7% at a dose of 2 mg/kg [63]. In mixed-breed dogs, the absolute oral bioavailability was 71.93 ± 9.75% at a dose of 5 mg/kg [64]. These data indicate that the oral bioavailability of LFX is high in various animal species including humans. Regarding the intestinal absorption mechanism of LFX, it was reported that the uptake of LFX by Caco-2 cells showed high- and low-affinity components with K(m) values of 0.489 and 14.6 mM, respectively, indicating that plural transporters are involved in the transport of LFX in Caco-2 cells. OATP1A2 was likely to function as a high-affinity transporter, and the active influx transport at least partially explains the high membrane permeability of the quinolone agents in various tissues [65]. In addition to the active influx transport of LFX, the contribution of P-glycoprotein (P-gp)-mediated efflux transport in the basolateral to apical transport in Caco-2 cells was also reported [66,67]. When absorptive versus secretive in vitro transport was compared among 4 quinolines, the contribution of efflux transport was in the following order: CFX > lomefloxacin > rhodamine 123 > LFX > OFX in Caco-2 cells, in which rhodamine 123 is one of typical P-gp substrates [68].

Regarding the chelate formation of LFX, many articles are available. For example, when the chelating ability was compared among cationic metals using four fluoroquinolones, LFX, CFX, sparfloxacin and enrofloxacin, the chelation potential of Al^3+^ was highest, whereas that of Mg^2+^ was lowest. It was also found that quinolones have a similar affinity for the metal ions, forming chelates which are more stable with hard Lewis acids such as the trivalent cations (Al^3+^, Fe^3+^) [69]. In quinolone chelation, divalent cations (Mg^2+^, Ca^2+^, Cu^2+^, Zn^2+^, Fe^2+^, Co^2+^, etc.) bind to carbonyl and carboxyl groups in neighboring positions and form chelates with 1:1 or 1:2 (metal: ligand) stoichiometry, and trivalent cations (A1^3+^, Fe^3+^) form chelates with 1:1, 1:2 or 1:3 (metal:ligand) stoichiometry [70]. Chelate formation using oxygen in a carboxylic acid at the 3-position and carbonyl oxygen atom at the 4-position of quinolones is reported in many articles [70,71,72,73]. Various ester prodrugs of LFX including LFX-EHE prevented chelate formation possibly by steric hindrance of the binding site in vitro and in vivo [31,32,33,74]. In addition to chelate formation in water, quinolones also interact with cationic metal-containing solid materials by physicochemical adsorption [15,29,75,76,77]. For example, the reported rate of adsorption of several quinolones (50 µM) onto solid Al(OH)_3_ (2.5 mg/mL) was in the following order: norfloxacin (NFX, 72.0%) > enoxacin (ENX, 61.0%) > LFX (48.1%) = OFX, (47.2%). In addition, the elution rate of adsorbed quinolones with water was in the following order: LFX (17.9%) approximately OFX (20.9%) approximately ENX (18.3%) > NLX (11.9%) [56]. These data would indicate that the adsorptive potency of LFX is relatively low among quinolones. In clinical, staggered dosing or separated administration by at least 2 h of LFX and antacids is required to avoid interaction with cationic metals [58,78].

In the present study, the oral bioavailability of LFX was 87.21 ± 5.59% in control rabbits and 119.04 ± 9.37% in EDTA-treated rabbits (Table 8), indicating that the oral bioavailability of LFX is reduced by gastric metals in control rabbits. Coadministration of Al(OH)_3_ reduced the oral bioavailability of LFX in both control and EDTA-treated rabbits. This would also indicate that EDTA 2Na used for treatment in the present study did not disturb chelate formation significantly. There was no difference in the oral bioavailability of LFX between LFX-EHE and LFX itself in both control and EDTA-treated rabbits (Table 8), although the lipophilicity of LFX-EHE was 13-fold higher than LFX [33]. The reason is not clear at present, but both LFX and LFX-EHE are absorbed efficiently. The oral bioavailability of LFX from LFX-EHE was comparable irrespective of the presence of Al(OH)_3_ in control and EDTA-treated rabbits, indicating that ester prodrugs of LFX can avoid chelate formation with Al^3+^. Although there was no difference in the oral bioavailability of LFX from LFX-EHE between the absence and presence of Al(OH)_3_, the average oral bioavailability of LFX in the presence of AL(OH)_3_ appeared to be reduced by 18.6% in the control rabbits and 28.5% in EDTA-treated rabbits, respectively (Table 8). This may be due to the adsorptive interaction between undissolved Al(OH)_3_ and LFX-EHE, because such interaction was not observed in the in vitro study (Table 2). Further study is necessary regarding the possibility of adsorptive interaction between ester prodrugs and cationic metals.

### 4.3. Regarding CFX and CFX-EHE

In BCS, CFX is classified as a Class IV drug with low solubility (approximately <1 mg/mL) and low permeability, and CFX hydrochloride is classifieds as a Class III drug with high solubility (approximately 36 mg/mL at 25 °C) and low permeability [79,80,81,82]. CFX is a zwitterionic compound with pKa = 6.18 (carboxylic acid group) and pKa = 8.76 (nitrogen on piperazinyl ring) [83], indicating the solubility/lipophilicity varies depending on the pH of the solution. In healthy male volunteers, the absolute oral bioavailability of CFX was approximately 70–85% [57,58,62,84,85,86]. For example, it was reported that the average absolute oral bioavailability of CFX (200 mg/kg) was 69% [59]. Another group reported that the oral bioavailability of CFX after administration of single oral doses of 100, 250, 500 and 1000 mg was 85%, and the 24-h urinary excretion was 42.2% after the 100-mg intravenous dose and 42.5% after the 500-mg twice-daily oral dose during steady state [87]. Separately, the absolute oral bioavailability of CFX in healthy human subjects at the two doses was similar: 69 and 69.1% for the 200 and 750 mg doses, respectively, in which the bioavailability with the 750 mg dose was significantly more variable [85]. Furthermore, the absolute bioavailability of CFX was approximately 70%, and fecal recovery of CFX accounted for approximately 15% of the dose [60]. In rats, the reported absolute oral bioavailability of CFX was 69.9% at a dose of 20 mg/kg [85]. In contrast, another group reported that the oral bioavailability of CFX was 19.87% when administered at a dose of 20 mg/kg in rats [88]. This reported value (19.87% in rats) is the same as the present results obtained in EDTA-treated rabbits (19.37 ± 2.28%) as shown in Table 9. In Caco-2 cells and rat jejunum, CFX exhibited polarized transport in the mucosal-to-serosal direction and CFX was a low-permeability drug and showed higher and pH-dependent transport in the mucosal-to-serosal direction than in the opposite direction [89]. Several fluoroquinolones are substrates of OATP1A2 expressed in the human intestine. Thus, to clarify the involvement of OATP in intestinal absorption of CFX, the contribution of OATP, or Oatp1a5, which is expressed at the apical membranes of rat enterocytes, to intestinal absorption of CFX was investigated using rat small intestine. The intestinal influx transport of CFX was affected by influx transporter Oatp1a5, in which the contribution of the ATP-dependent efflux transporters, such as P-gp and breast cancer resistant protein (Bcrp), was also observed [90]. It was also reported that MDCKI and MDCKI-MDR1 cells indicated that CFX is a substrate of P-gp but MDCKII, MDCKII-MDR1, LLC-PK1 and L-MDR1 cells suggested that CFX may not be a substrate of P-gp. Based on these data, it was commented that further studies are necessary to characterize these P-gp overexpressing cell lines and the transport mechanism of CFX [91]. Other research groups reported that the main mechanism of CFX transport through Caco-2 cells in both directions is active diffusion and P-gp [92].

Regarding the interaction of CFX with cationic metals, many articles are available. For example, it was reported that the dissolution rate of CFX-containing tablets was markedly retarded in the presence of antacids, such as sodium bicarbonate, calcium hydroxide, calcium carbonate, Al(OH)_3_, Mg(OH)_2_, magnesium carbonate, magnesium trisilicate and magaldrate, resulting in decreased oral bioavailability of CFX [19,93,94,95]. As well, cationic metal-containing materials such as sucralfate, Ca-fortified orange juice, milk, dairy products containing cationic metals and so on also significantly decreased the absorption of CFX [96,97,98]. Similarly, in the interaction of CFX with cationic medals, the physicochemical adsorption to polyvalent metal cation products is also involved, in addition to chelate formation [98]. For example, it was reported that the concomitant intake of CFX film-coated tablets and milk decreased the absorbable amount of CFX not only by chelation but also by adsorption of CFX on the surface of proteins [97]. As well as in the case of LFX, in the case of quinolones, cationic metal binds to one oxygen of a carboxylic acid and ring carbonyl oxygen of CFX [69,70,71,72]. Thus, the addition of ester moiety at the carboxyl acid of CFX was thought to inhibit the binding of metal cations by steric hindrance. Many other ester prodrugs of CFX are also expected to inhibit the chelate formation of CFX as observed with LFX ester prodrugs and CFX-EHE [32,33,70,74]. Regarding the adsorptive interaction of CFX, it was reported that when various antibiotics were incubated in soil-water slurries (silt-loam, sand-loam, and sand; 20 *w*/*v* % soil), β-lactams and florfenicol antibiotics remain bioactive in soils while CFX, neomycin, and TC were neutralized [52]. Among various quinolones, CFX is thought to be one of the representative adsorptive antibiotics, as well as TC, and many articles are available regarding the adsorptive interaction of CFX with metal-containing solid materials.

In the present study, in rabbits, the absolute oral bioavailability of CFX was 13.73 ± 2.48% in control rabbits and 19.37 ± 2.28% in EDTA-treated rabbits, in which EDTA-treated rabbits exhibited significantly higher oral bioavailability of CFX than control rabbits. Coadministration of Al(OH)_3_ decreased the oral bioavailability of CFX in both control and EDTA-treated rabbits (Table 9). In contrast, as well as in the case of LFX-EHE, the oral bioavailability of CFX from CFX-EHE was not significantly reduced by the presence of Al(OH)_3_ in control and EDTA-treated rabbits, indicating that ester prodrugs of CFX can avoid chelate formation with Al^3+^ in vitro and in vivo (Table 2 and Table 9). There was no difference in the oral bioavailability of CFX between CFX-EHE and CFX itself in both control and EDTA-treated rabbits, irrespective of the 13-fold higher lipophilicity (PC) of CFX-EHE than CFX itself (Table 1 and Table 9). In this case, CFX-EHE is thought to be absorbed by passive diffusion, but CFX itself is absorbed mainly by OATP-mediated transport as discussed already. Further study is necessary to clarify the mechanism for the no difference in the oral bioavailability of CFX between CFX and CFX-EHE. In addition, although there was no significant difference, the presence of Al(OH)_3_ appeared to decrease the bioavailability of CFX from CFX-EHE, possibly by the adsorptive interaction between CFX-EHE and solid Al(OH)_3_ and/or substrate specificities of CFX-EHE for P-gp. Further study is necessary to clarify the mechanism of the scattering oral bioavailability of CFX from CFX-EHE in the presence of Al(OH)_3_.

### 4.4. Regarding TC

The absolute oral bioavailability of TC, a chelating/adsorptive drug, in humans, is reported to be 77–88% at a dose of 250 mg, in which 5% of TC was metabolized to a less active metabolite and the remainder was eliminated by renal excretion [99,100]. In rats, TC is subjected to enterohepatic circulation, and it was reported that the amounts of TC absorbed from the bile and aqueous solutions were 72.92 and 77.34%, respectively, during 4 h. Accordingly, it was reported that biliary excretion of TC does not account for a significant elimination of this antibiotic from the body [101]. Separately, it was reported that the oral bioavailability of TC was almost identical from an oral solution and from capsules in fasted subjects, and the bioavailability of TC was decreased by approximately 50% by taking foods [102]. In addition, TC bioavailability was markedly decreased when TC was co-administered with antacids, milk, food containing Ca and so on [103,104,105,106,107,108]. Tetracyclines have a high affinity to form chelates with polyvalent metallic cations such as Fe^3^, Fe^2+^, Al^3+^, Mg^2+^ and Ca^2+^. Many of these tetracycline-metal complexes are either insoluble or otherwise poorly absorbable from the gastrointestinal tract [109].

When analyzed with circular dichroism (CD) spectrum, Ca^2+^ was found to form a 2:1 metal-ion complex with TC, and Mg^2+^ formed the complex at a 1:1 ratio. In the Ca^2+^-TC complex, Ca^2+^ binds to the C-10 and C-11 sites of TC with the subsequent addition of a second metal ion at the C-12 and C-1 sites. The Mg^2+^ chelation occurred at the C-11 and C-12 β-diketone sites of TC [110,111,112]. In addition to the chelate formation of TC in water, TC interacts with various cationic metals by physiochemical adsorption [20,21,23,24,25,26,27]. The adsorption of TC to metal-containing materials is pH-dependent and can be varied depending on the concentrations of metal ions. TC has three pKa values, and at a pH lower than pKa_1_ (3.3), the TC was dominant with species of TC^3+^. With the pH ranging between pKa_1_ and pKa_2_ (3.3 < pH < 7.7), the TC was presented with TC^2+^. At pH further increasing to a range of (7.7 < pH < 9.7) and (pH > 9.7), the dominant species of TC were TC^−^ and TC^2−^, respectively [113]. Maximum TC adsorption on ferrihydrite occurred at pH 5–6, and the TC adsorption was promoted by the addition of Cu^2+^, Zn^2+^ and Al^3+^. Separately, the adsorption process on discarded natural coal gangue was classified as a thermodynamic endothermic and spontaneous reaction, which was controlled by chemical and physical adsorption, including electrostatic interaction between negatively charged TC and cationic metals and cation exchange between positively charged TC and cationic metals, in addition to physical adsorption [23,114]. Like this, TC contains more functional groups than CFX for binding, resulting in greater adsorption affinity [53]. In clinical, TC is administered at a dose of 250 mg or 500 mg in human adults, which would correspond to less than 10 mg/kg. In the previously reported article, however, TC was administered at a dose of 150 mg/kg in unfasted and fasted rabbits, in which, in the case of fasted rabbits, rabbit food (Purina Rabbit Chow) was withdrawn for 12 h prior to TC administration. Values of Tmax and AUC of TC in fasted rabbits were greater than those in non-fasted rabbits [115].

In the present study, TC was also administered at a dose of 150 mg/kg, because TC was not detected in plasma at a lower dose than 30 mg/kg. The absolute oral bioavailability of TC in control rabbits was 2.56 ± 0.95%, and it significantly increased to 4.90 ± 0.81% in EDTA-treated rabbits. These oral bioavailabilities (<5% of dose) of TC in rabbits, however, are far lower than those reported (77–88% at a dose of 250 mg) in humans [99,100]. The reason for the marked difference in TC bioavailability between humans and rabbits is not known at present. However, the detectable absorption of TC at the higher dose may be due to the saturation of adsorption in the gastrointestinal lumen of rabbits. In a separate experiment, 1 mg/mL EDTA 2Na was ingested twice (one day before and 1 h before the experiment, >20 mL of 1 mg/mL EDTA 2Na including drinking water during one day) before experiments, and then the bioavailability of TC was determined at a dose of 150 mg/kg. There was no significant difference in TC plasma concentrations between 10 mL and 20 mL ingestion of 1 mg/kg EDTA 2Na, possibly the cationic metals in gastric water were almost eliminated by the ingestion of 10 mL of 1 mg/mL EDTA 2Na (Table 5). Further study is necessary to clarify the reason for the quite low oral bioavailability of TC even in EDTA-treated rabbits, as compared with the bioavailability in humans (Table 5).

### 4.5. Comparison of Oral Bioavailabilities of LFX, CFX, and TC between Rabbits and Humans

The oral bioavailabilities of LFX, CFX and TC were compared between humans, control rabbits, and EDTA-treated rabbits (Table 11).

The oral bioavailability of LFX was comparable between humans and rabbits (control and EDTA-treated rabbits). In contrast, the oral bioavailability of CFX and TC, especially TC, in rabbits were markedly lower as compared with those in humans, The chelating ability of these antibiotics to Al^3+^ in water was in the following order: TC > CFX > FLX (Table 2). However, in the case of EDTA-treated rabbits, the chelating interaction of each antibiotic with gastric metals is considered to be small if any because of the low gastric metal concentrations (Table 5). The adsorptive potencies of each antibiotic to cationic metal-containing materials are thought to be in the following order: TC > CFX > FLX, based on the literature as discussed in the Discussion section [53,56,58,78]. Taken together, it may be speculated that the large inter-species differences in oral bioavailability of TC and CFX, but not LFX, between humans and EDTA-treated rabbits may come from the difference in physicochemical adsorption potencies among them. TC and CFX are known to be representative antibiotics having potent adsorptive potencies among various chelating drugs [20,21,23,24,25,26,27,53,56,58,78], It will be important to clarify the contribution of physicochemical adsorption in the interspecies difference of oral bioavailability using TC, CFX and EDTA-treated rabbits.

## 5. Conclusions

EDTA-treated rabbits with low amounts of gastric contents including metals and low gastric pH were developed without causing mucosal damage as experimental animals for preclinical bioavailability studies. The treatment of rabbits with >10 mL of 1 mg/mg EDTA 2Na decreased the gastric metal contents to less than 1% of untreated control rabbits and gastric pH to physiologically normal levels. The oral bioavailabilities of LFX, CFX and TC in EDTA-treated rabbits were significantly higher than those in control rabbits, indicating that, in general, the oral bioavailabilities of chelating drugs are reduced by the interaction with gastric metals in rabbits. Ester prodrugs of fluoroquinolones such as LFX-EHE and CFX-EHE were effective in preventing chelate formation in gastrointestinal water but may not be effective in preventing physicochemical adsorption to metal-containing materials, because the oral bioavailabilities of TC and CFX, which are both potent adsorptive antibiotics to metal-containing materials, in EDTA-treated rabbits were far lower as compared with reported oral bioavailabilities. In this study, it was found that EDTA-treated rabbits exhibit quite low gastric contents, low gastric metals and low gastric pH, as well as fasted rats and humans. Thus, EDTA-treated rabbits may be expected to be useful preclinical experimental animal models in studying various drugs and dosage formulations. Further study is necessary to empty the gastrointestinal contents of EDTA-treated rabbits completely, to clarify the contribution of adsorptive interaction with gastrointestinal metals, and to seek out the usefulness of the EDTA-treated rabbit as an experimental animal model.

## Figures and Tables

**Figure 1 pharmaceutics-15-01589-f001:**
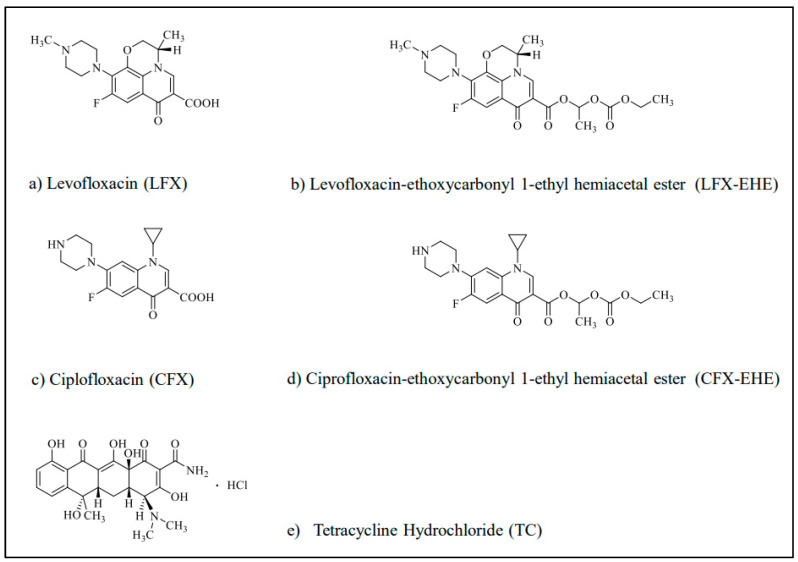
Chemical structures of (**a**) levofloxacin (CFX), (**b**) levofloxacin-ethoxycarbonyl 1-ethyl hemiacetal ester (LFX-EHE), (**c**) ciprofloxacin (CFX), (**d**) ciprofloxacin-methoxycarbonyl 1-ethyl hemiacetal ester (CFX-EHE), and (**e**) tetracycline hydrochloride used in the present study.

**Figure 2 pharmaceutics-15-01589-f002:**
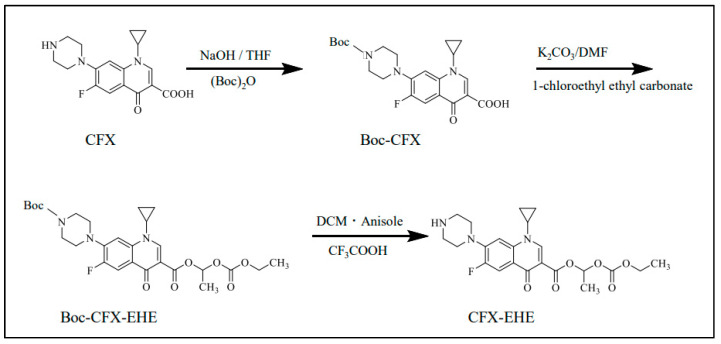
Stepwise scheme of the synthesis process of CFX-EHE from CFX via Boc-CFX and Boc-CFX-EHE.

**Figure 3 pharmaceutics-15-01589-f003:**
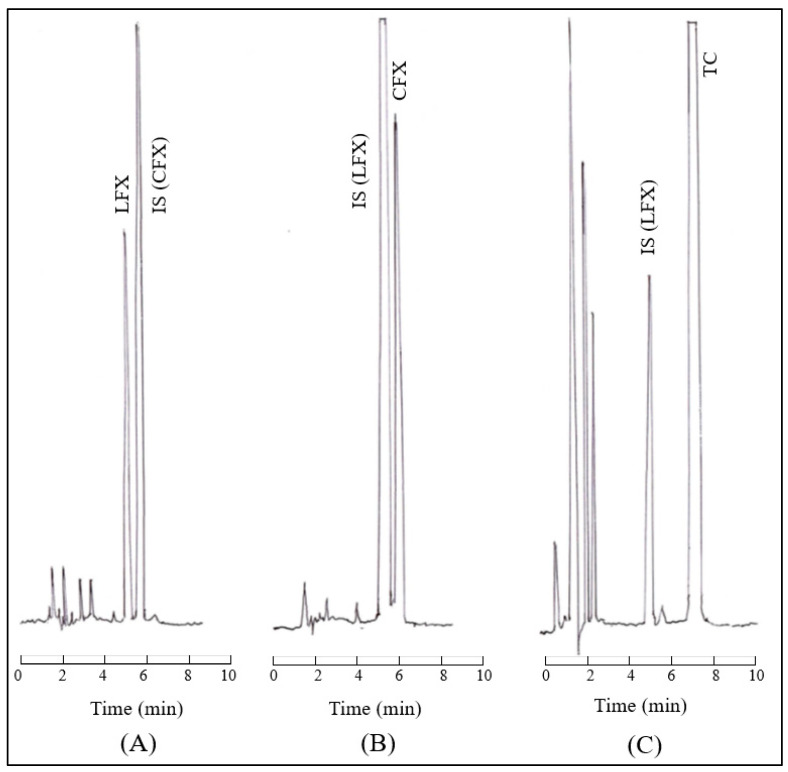
Chromatograms of (**A**) LFX (0.5 μg/mL) and CFX (IS), (**B**) CFX (0.5 μg/mL) and LFX (IS), and (**C**) TC (50 μg/mL) and LFX (IS) in control rabbits’ plasma.

**Figure 4 pharmaceutics-15-01589-f004:**
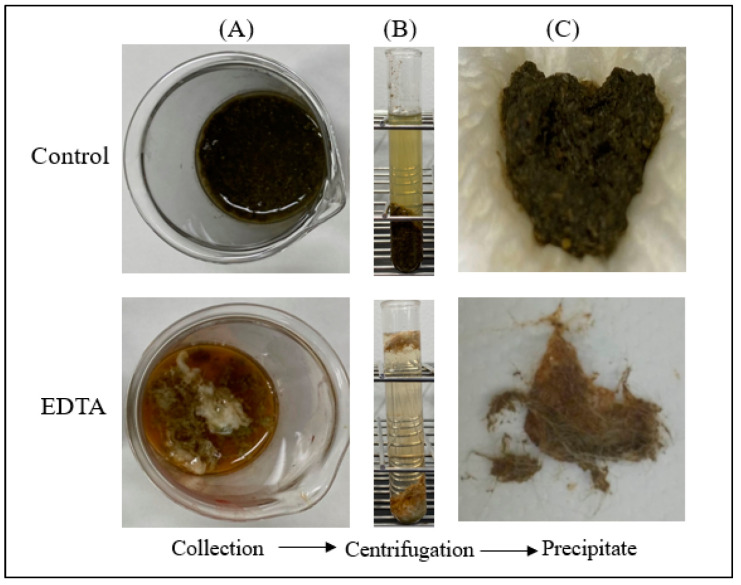
Comparison of gastric contents obtained from control and EDTA-treated rabbits. Whole gastric contents were collected in a beaker and weighed (**A**). The suspended contents were centrifuged to measure the water volume in gastric contents (**B**). Precipitates of gastric contents of control and EDTA-treated rabbits (**C**). Precipitates of control rabbits contained feces, food pellets and hair, and EDTA-treated rabbits contained hair and undisintegrated food pellets.

**Figure 5 pharmaceutics-15-01589-f005:**
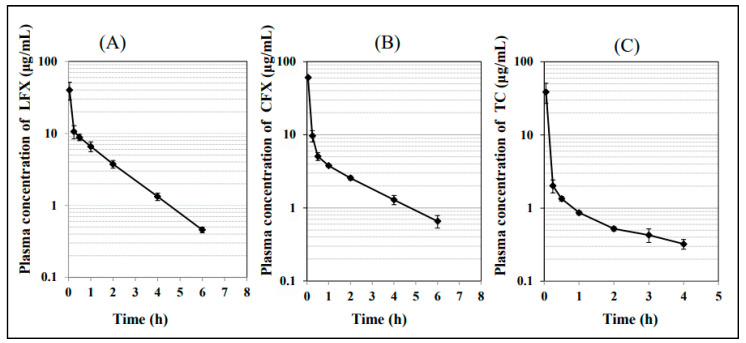
Plasma disposition of LFX (**A**), CFX (**B**), and TC (**C**) after a 2.8-min intravenous infusion in control rabbits. The doses of LFX, CFX, and TC for intravenous infusion were 20 mg/kg, 30 mg/kg and 10 mg/kg, respectively. Each value represents the mean ± SD (*n* = 3).

**Figure 6 pharmaceutics-15-01589-f006:**
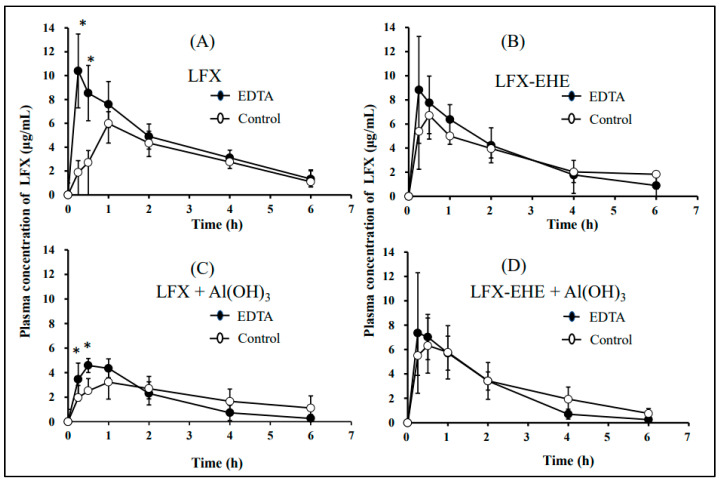
Plasma concentration-time profiles of LFX after oral administration of LFX alone (**A**), LFX and Al(OH)_3_ (**C**), LFX-EHE alone (**B**), and LFX-EHE and Al(OH)_3_ (**D**) in control (open circle) and EDTA treated rabbits (closed circle). The doses of LFX and LFX-EHE were 20 mg/kg as CFX, and the dose of Al(OH)_3_ was 100 mg/kg. Each value represents the mean ± SD (*n* = 3). * Significantly different from LFX in control rabbits at a level of *p* < 0.05.

**Figure 7 pharmaceutics-15-01589-f007:**
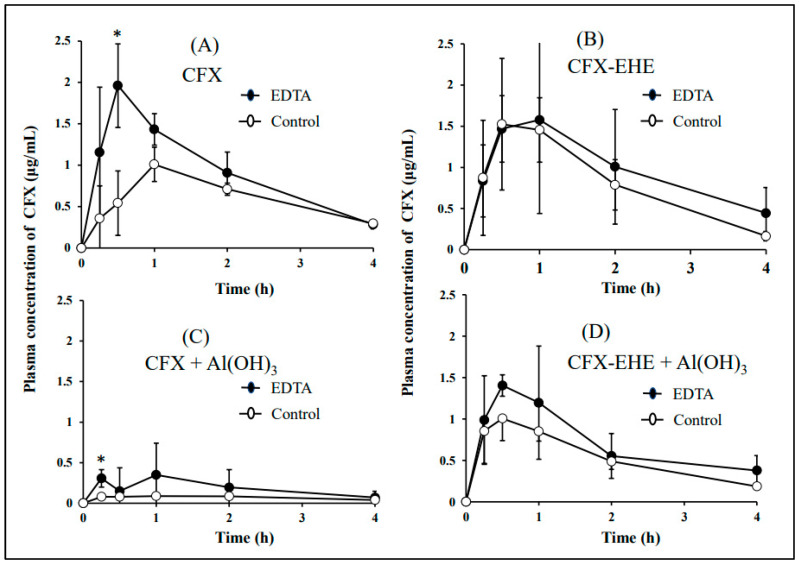
Plasma concentration-time profiles of CFX after oral administration of CFX alone (**A**), CFX and Al(OH)_3_ (**C**), CFX-EHE alone (**B**), and CFX-EHE and Al(OH)_3_ (**D**) in control (open circle) and EDTA treated rabbits (closed circle). The doses of CFX and CFX-EHE were 30 mg/kg as CFX, and the dose of Al(OH)_3_ was 100 mg/kg. Each value represents the mean ± SD (*n* = 3). * Significantly different from CFX in control rabbits at a level of *p* < 0.05.

**Figure 8 pharmaceutics-15-01589-f008:**
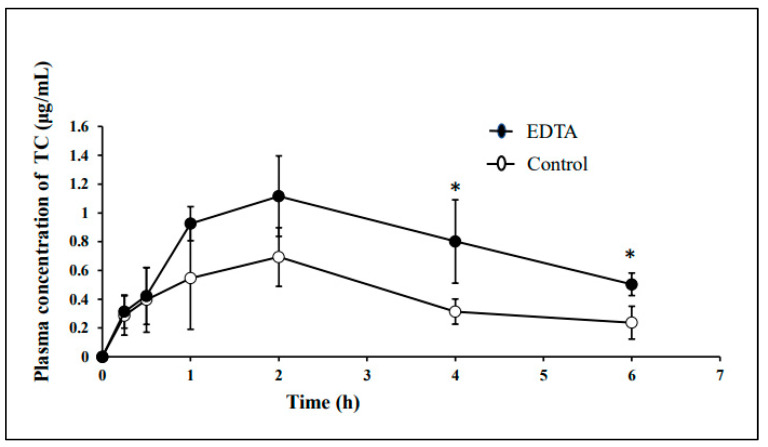
Plasma concentration–time profiles of TC after oral administration at a dose of 150 mg/kg in control (open circle) and EDTA-treated rabbits (closed circle). Each value represents the mean ± SD (*n* = 3). * Significantly different from TC in control rabbits at a level of *p* < 0.05.

**Table 1 pharmaceutics-15-01589-t001:** Lipophilicity and solubility of CFX and CFX-EHE.

	Lipophilicity			Solubility (mg/mL)	
Compound	PC	log PC	Clog *p*	pH 2.0	pH 6.5
CFX	0.46	−0.34	1.32	8.82	0.71
CFX-EHE	6.12	0.79	2.74	4.28	3.08

PC: Partition coefficient was determined in a chloroform/0.1 M Tris-HCl buffer (pH 6.5) partition system at 25 °C. Calculated log *p* was estimated using ChemDraw, a chemical drawing tool. Solubility was determined using pH 2 buffer adjusted with hydrochloric acid and 0.1 M Tris-HCl buffer (pH 6.5) at 25 °C. Each value represents the mean of 3 trials.

**Table 2 pharmaceutics-15-01589-t002:** The precipitation percentages of drugs when mixed with various concentrations of AlCl_3_ at pH 6.5.

	Precipitation (%)				
Compound/AlCl_3_	25 μM	50 μM	100 μM	500 μM	1 mM
CFX	20.4 ± 17.6	22.9 ± 16.7	39.2 ± 17.7	55.5 ± 18.0	73.4 ± 2.0
CFX-EHE	0	0	0	0	0
TC	63.2 ± 21.3	84.0 ± 13.9	91.6 ± 7.3	96.2 ± 5.7	99.4 ± 1.19

Each drug and Al(OH)_3_ were dissolved in 0.5 mM Tris-HCl buffer (pH 6.5) and mixed in a 1:1 *v*/*v* ratio. The final concentration of a drug was 50 μM and that of AlCl_3_ was 25 μM, 50 μM, 100 μM, 500 μM or 1 mM. The mixture solution was stood for 30 min at 24 °C. The value of 0 indicates that no precipitation occurred. Each value represents the mean ± SD (*n* = 3).

**Table 3 pharmaceutics-15-01589-t003:** Stability of CFX-EHE in buffer solution, and enzymatic solution at 37 °C.

	Hydrolysis (%)		
Medium	1 min	5 min	15 min
0.1 M PBS (pH 6.5)	0	0	0
0.1 M PBS (pH 7.4)	0	0	0
2% Mucosal homogenate	20.4 ± 2.7	59.0 ± 9.2	85.9 ± 11.5
2% Liver homogenates	38.4 ± 15.8	69.5 ± 9.5	94.8 ± 3.4
10% Plasma	9.5 ± 9.0	27.2 ± 1.0	71.1 ± 17.4

Membrane mucosal surface of the small intestine, liver and plasma were obtained from rats, and each biological specimen was prepared using pH 7.4, 25 mM Tris-HCl buffer. Each value represents the mean ± SD (*n* = 3).

**Table 4 pharmaceutics-15-01589-t004:** Minimum inhibitory concentrations (MICs) of CFX and CFX-EHE.

Microorganism		MICs (µg/mL)	
		CFX	CFX-EHE
*S. aureus*	ATCC29213	0.25	4.00
*E. coli*	ATCC25922	0.008	0.50
*P. aeruginosa*	ATCC27853	0.50	8.00

MIC (minimum inhibitory concentrations) was determined by agar plate dilution method using Mueller–Hinton broth. Each value represents the mean of three trials.

**Table 5 pharmaceutics-15-01589-t005:** Gastric contents of untreated control, control and EDTA-treated rabbits.

Rabbit Group	Gastric Contents (g)	Water Contents (mL)	Gastric pH	Concentrations	
				Ca (mg/dL)	Mg (mg/dL)
Untreated control	(1) 82.2 (feces > pellets > hair)	50.0	2.6	159.5	22.8
	(2) 72.2 (feces > pellets > hair)	45.0	3.7	78.7	12.8
Control	(1) 43.3 (feces > pellets > hair)	13.5	1.5	13.6	2.4
	(2) 14.4 (feces > hair)	9.8	1.4	68.9	11.0
EDTA-treated	(1) 9.5 (hair > pellets)	6.7	1.6	2.6	1.9
	(2) 7.8 (hair > pellets)	4.8	1.2	1.3	1.6
	(3) 13.1 (hair > pellets)	10.5	1.3	1.9	1.9

Untreated control rabbits were kept in steel cages and had free access to food and water and coprophagy; control rabbits were kept in steel cages and fasted with free access to water one day before the experiment. Coprophagy was not prevented. Ten mL of water was administered orally at 1 h before experiments; EDTA-treated rabbits were kept in fixing boxes to prevent food intake and coprophagy, but free access to 1 mg/mL EDTA 2Na solution. Ten mL of 1 mg/mL EDTA 2Na solution was administered orally at 1 h before experiments.

**Table 6 pharmaceutics-15-01589-t006:** Oral bioavailability of FD-4 in control and EDTA-treated rabbits.

	Plasma FD-4 Concentrations (µg/mL)			
Rabbits	0.5 h	1.0 h	2.0 h	4.0 h
Control	0.59 ± 0.31	0.42 ± 0.30	0.29 ± 0.29	0.28 ± 0.27
EDTA-treated	0.35 ± 0.30	0.41 ± 0.30	0.27 ± 0.29	0.24 ± 0.08

The dose of FD-4 was 10 mg/kg. Each value represents the mean ± SD (*n* = 3).

**Table 7 pharmaceutics-15-01589-t007:** Pharmacokinetic parameters of LFX, CFX and TC after a 2.8-min intravenous infusion in control rabbits.

Drug (Dose)	Cmax (µg/mL)	β (h^−1^)	AUC_0–6h_ (µg h/mL)	AUC_0–∞_ (µg h/mL)
LFX (20 mg/kg)	40.21 ± 11.34	0.53 ± 2.91	23.42 ± 2.91	24.29 ± 2.83
CFX (30 mg/kg)	60.74 ± 1.92	0.34 ± 0.04	19.64 ± 0.48	23.56 ± 0.74
TC (10 mg/kg)	38.83 ± 11.86	0.28 ± 0.10	7.57 ± 1.38 (a)	8.85 ± 1.67

(a) AUC_0–4h_. Each value represents the mean ± SD (*n* = 3).

**Table 8 pharmaceutics-15-01589-t008:** Pharmacokinetic parameters of LFX after oral administration of LFX and LFX-EHE with or without Al(OH)_3_ in control and EDTA-treated rabbits.

		Cmax (μg/mL)	Tmax (h)	AUC_0–6h_ (μg h/mL)	AUC_0–∞_ (μg h/mL)	Bioavailability (%)
Rabbits	Dosing					
Control	LFX	6.41 ± 0.90	1.33 ± 0.58	19.11 ± 0.57	21.18 ± 1.36	87.21 ± 5.59
	+Al(OH)_3_	3.39 ± 1.18 *	1.17 ± 0.76	12.35 ± 3.20 *	14.45 ± 5.03	59.52 ± 20.70
EDTA-treated	LFX	11.32 ± 2.59 *	0.33 ± 0.14 *	26.38 ± 3.45 *	28.91 ± 2.28 *	119.04 ± 9.37 *
	+Al(OH)_3_	5.00 ± 0.26	0.67 ± 0.29	11.04 ± 3.05	11.55 ± 3.64 *	47.57 ± 15.01 *
Control	LFX-EHX	8.86 ± 3.97	0.42 ± 0.14	21.52 ± 6.14	23.37 ± 6.19	96.29 ± 5.82
	Al(OH)_3_	6.80 ± 1.98	0.42 ± 0.14	17.84 ± 7.24	19.27 ± 7.9	79.33 ± 32.88
EDTA-treated	LFX-EHE	9.36 ± 3.76	0.58 ± 0.38	20.71 ± 1.20	22.41 ± 1.41	92.29 ± 5.82
	Al(OH)_3_	8.58 ± 3.59	0.33 ± 0.14 *	15.55 ± 3.43	16.03 ± 3.55	66.01 ± 11.63

The oral dose of LFX and LFX-EHE was 20 mg/kg as LFX, and the dose of Al(OH)_3_ was 100 mg/kg. Each value represents the mean ± SD. (*n* = 3). Absolute oral bioavailability was estimated by comparing AUC_0–∞_ after oral and intravenous administrations of LFX (Table 7). * Significantly different from LFX in control rabbits at a level of *p* < 0.05.

**Table 9 pharmaceutics-15-01589-t009:** Pharmacokinetic parameters of CFX after oral administration of CFX and CFX-EHE with or without Al(OH)_3_ in control and EDTA-treated rabbits.

		Cmax (μg/mL)	Tmax (h)	AUC_0–6h_ (μg h/mL)	AUC_0–∞_ (μg h/mL)	Bioavailability (%)
Rabbits	Dosing					
Control	CFX	1.01 ± 0.21	1.00 ± 0.00	2.37 ± 0.56	3.24 ± 0.59	13.73 ± 2.48
	+Al(OH)_3_	0.12 ± 0.01 *	1.08 ± 0.88	0.28 ± 0.07 *	0.40 ± 0.12 *	1.70 ± 0.53 *
EDTA-treated	CFX	1.96 ± 0.50 *	0.50 ± 0.00	3.73 ± 0.42 *	4.56 ± 0.54 *	19.37 ± 2.28 *
	+Al(OH)_3_	0.48 ± 0.23 *	0.50 ± 0.43	0.75 ± 0.43 *	0.96 ± 0.45 *	4.06 ± 1.92 *
Control	CFX-EHX	1.67 ± 0.68	0.58 ± 0.38	3.23 ± 0.84	3.71 ± 0.90	15.76 ± 3.83
	Al(OH)_3_	1.06 ± 0.22	0.67 ± 0.29	2.15 ± 0.17	2.69 ± 0.16	11.44 ± 0.67
EDTA-treated	CFX-EHE	2.00 ± 0.73	0.67 ± 0.29	3.90 ± 2.03	5.20 ± 2.66	22.09 ± 11.31
	Al(OH)_3_	1.59 ± 0.29	0.58 ± 0.38	2.83 ± 1.08	3.94 ± 1.58	16.71 ± 6.71

The oral dose of CFX and CFX-EHE was 30 mg/kg as CFX, and the dose of Al(OH)_3_ was 100 mg/kg. Each value represents the mean ± SD (*n* = 3). Absolute oral bioavailability was estimated by comparing AUC_0–∞_ after oral and intravenous administrations of CFX (Table 7). * Significantly different from CFX in control rabbits at a level of *p* < 0.05.

**Table 10 pharmaceutics-15-01589-t010:** Pharmacokinetic parameters of TC (150 mg/kg) given orally in control and EDTA-treated rabbits.

		Cmax (μg/mL)	Tmax (h)	AUC_0–6h_ (μg h/mL)	AUC_0–∞_ (μg h/mL)	Bioavailability (%)
Rabbits	Dosing					
Control	TC HCl	0.73 ± 0.22	1.67 ± 0.58	2.54 ± 0.91	3.39 ± 1.26	2.56 ± 0.95
EDTA-treated	TC HCl	1.12 ± 0.28	2.00 ± 0.00	4.68 ± 0.93 *	6.49 ± 1.08 *	4.90 ± 0.81 *

Absolute oral bioavailability was estimated by comparing AUC_0–∞_ after oral and intravenous administrations of CFX (Table 7). Each value represents the mean ± SD. (*n* = 3). * Significantly different from TC in control rabbits at a level of *p* < 0.05.

**Table 11 pharmaceutics-15-01589-t011:** Oral bioavailabilities of LFX, CFX and TC in humans, control rabbits and EDTA-treated rabbits.

	LFX	CFX	TC
Humans	70–100% (200–750 mg)	70–85% (200–1000 mg)	77–88% (250 mg)
Control rabbits	87.2 ± 5.6% (20 mg/kg)	13.7 ± 2.5% (30 mg/kg)	2.5 ± 1.0% (150 mg/kg)
EDTA-treated rabbit	119.0 ± 9.4% (20 mg/kg)	19.4 ± 2.3% (30 mg/kg)	4.9 ± 0.8% (150 mg/kg)

The values in parentheses represent the dose. The oral bioavailabilities in humans were cited from references (listed in the discussion section) and those in rabbits are the same as those in Table 5. The oral bioavailabilities in EDTA-treated rabbits are significantly higher than those in control rabbits.

## Data Availability

Not applicable.

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
