# Peer review of "Development and Evaluation of EDTA-Treated Rabbits for Bioavailability Study of Chelating Drugs Using Levofloxacin, Ciprofloxacin, Hemiacetal Ester Prodrugs, and Tetracycline"

_pharmaceutics, 2023, doi:10.3390/pharmaceutics15061589_

Round 1

Reviewer 1 Report

This is a good manuscript.  To me the manuscript is scientifically sound.  I hav only two minor comments.

Line 275: "Blood ssamples .." should resd "Blood samples ...".

Line 315: "=In ..." should read "In ...".

Author Response

Thank you for review of our manuscript.

Line 275: "Blood ssamples .." should resd "Blood samples ...".  

         It was changed to samples

Line 315: "=In ..." should read "In ...".    

          “=” was delated.

Reviewer 2 Report

1.The title of this manuscript are hemiacetal ester prodrugs, while the key word is ester prodrug, so it is suggested to modify;

 2. edta 2Na can stimulate the mucosa and upper respiratory tract. Will oral pretreatment of edta 2Na affect the mucosa and upper respiratory tract of rabbits? How to determine the amount of medicine? Please supplement the argument.

3.In part 2.9.1 of the manuscript, it is mentioned that "some contents separate the heart at 3000 RPM for 10 minutes", the term of this part is not clear, the specific number of milliliters? Please add.

4.The table layout in the paper is wrong, such as table1, table4, table8, table9, table10, it is suggested to modify;

5.Al(OH)3 in 3.4.3 is white gelatinous precipitate, which will react with CFX to form chelates. May I ask whether the "co-administration" mentioned in the article is the simultaneous administration or the sequence? Please be specific.

6. In 3.4.4, it said, "Please explain the purpose of oral administration of 20ml edta 2Na(10ml the day before the experiment and 10ml on the day of the experiment);

7.In Discussion 6.4, relevant knowledge about another experiment is mentioned, however there no literature is cited. It is suggested to add.

8. The format of the reference is incorrect. For example, the font color of 20, 101, 102, 104, 105 is not consistent with the context; The spacing of the reference are not uniform.ï¼›Document type identification should be indicated after the title; The author should to check the references in detail;

9. P values in the paper are not uniform, some are not marked with asterisks, some are not italic and not capitalized, such as table8 and table10, it is suggested to modify;

10.The font size is not uniform. For example, the 888 line is small and inconsistent with the context.

Author Response

Thank you for your review

1.The title of this manuscript are hemiacetal ester prodrugs, while the key word is ester prodrug, so it is suggested to modify;     The keyword was changed to hemiacetal ester prodrugs

  1. edta 2Na can stimulate the mucosa and upper respiratory tract. Will oral pretreatment of edta 2Na affect the mucosa and upper respiratory tract of rabbits? How to determine the amount of medicine? Please supplement the argument.

The stimulating effect of 1 mg/mL EDTA 2Na solution on the mucosa and upper respiratory tract is thought to be low if any, because, in the present study, the dosing of 10 mL EDTA solution was made by using stomach tube. In the case of drinking (EDTA solution), the solution will pass the throat, but not be retained there, different from the case of the rectal administration. EDTA 2Na is a hydrophilic compound, which may imply that the absorption of EDTA through mucous layer is low. In addition, EDTA is thought to be very low toxic compounds, because in EDTA chelation therapy in clinical, EDTA 2Na or EDTA-Ca is infused at a dose of 3 g for 1.5 – 3 hrs, which would suggest that EDTA 2Na is a relatively low toxic compound (Seeley, et al., EDTA chelation therapy for cardiovascular disease: a systematic review. BMC Cardiovascular Disorders, 2005; 5: Article number: 32). 

  1. In part 2.9.1 of the manuscript, it is mentioned that "some contents separate the heart at 3000 RPM for 10 minutes", the term of this part is not clear, the specific number of milliliters? Please add.

I am sorry, but I couldn’t find the corresponding sentence. In part 2.9.1. it is described that “The stomach contents were taken out into a beaker and weighed, and a part of the contents was centrifuged at 3,000 rpm for 10 min”. The sentence of “a part of the contents was centrifuged at 3,000 rpm for 10 min” was changed to “the suspension was centrifuged at 3,000 rpm for 10 min to measure the amount of gastric water”

4.The table layout in the paper is wrong, such as table1, table4, table8, table9, table10, it is suggested to modify;  

The layout of tables were modified.  

5.Al(OH)3 in 3.4.3 is white gelatinous precipitate, which will react with CFX to form chelates. May I ask whether the "co-administration" mentioned in the article is the simultaneous administration or the sequence? Please be specific.  

Drug solutions and Al(OH)3 suspension were administered separately as follows: Administration of Al(OH)3 suspension (approximately 3mL) and water (approximately 2 mL) to wash the tube, and then drug solution (approximately 3 mL) followed by water (approximately 2 mL). The total amount of water was 10 mL.  This was added to the experimental section (2.9.3)

  1. In 3.4.4, it said, "Please explain the purpose of oral administration of 20ml edta 2Na(10ml the day before the experiment and 10ml on the day of the experiment);  

The higher dose of EDTA 2Na can chelate higher amount of cationic metals. However, there was no difference in TC absorption between 10 mL and 20 mL EDTA 2Na, as described in 3.4.4. section. As shown in Table 5, the almost of gastric metals were eliminated from the stomach by 10 mL of EDTA 2Na.

  1. In Discussion 6.4, relevant knowledge about another experiment is mentioned, however there no literature is cited. It is suggested to add.

To my knowledge, there are no closely related reports regarding the effect of gastric metals on the oral bioavailability of chelating drugs. Also, there is no report describing the treatment of rabbits with EDTA 2Na to eliminate gastric metals so far.

  1. The format of the reference is incorrect. For example, the font color of 20, 101, 102, 104, 105 is not consistent with the context; The spacing of the reference are not uniform.ï¼›Document type identification should be indicated after the title; The author should to check the references in detail;

The format of references was corrected.

  1. P values in the paper are not uniform, some are not marked with asterisks, some are not italic and not capitalized, such as table8 and table10, it is suggested to modify;

The expression of P was corrected (Tables 8-10).

10.The font size is not uniform. For example, the 888 line is small and inconsistent with the context.

The correction of font size was made.

Reviewer 3 Report

Overall, the MS is well written. I only have a few minor suggestions. 

1.    Typo error: Line 543, please correct Al(OH)3 to Al(OH)3

2.       Figure 5-7, why the author plot EDTA treated rabbit dots with the error bar on top and control rabbits in bottom error bar? I think the author should provide either the top or bottom error bar.

3.       Table 11, the table format has been changed, please correct.

Author Response

Thank you for your review.

  1. Typo error: Line 543, please correct Al(OH)3 to Al(OH)3

   Correction was made.

  1. Figure 5-7, why the author plot EDTA treated rabbit dots with the error bar on top and control rabbits in bottom error bar? I think the author should provide either the top or bottom error bar.

 The correction was made: error bars were added to the top and bottom because the addition of error bars didn’t disturb the plasma concentration values. When there is a significant difference between the two values, the mark of statistical significance (*) was also added.

  1. Table 11, the table format has been changed, please correct.

Format of table was corrected.

Reviewer 4 Report

Development and evaluation of EDTA-treated rabbits for bioavailability study of chelating drugs using levofloxacin, ciprofloxacin, hemiacetal ester prodrugs, and tetracycline is the main question addressed by the research. The topic is very interesting and research is novel. I belive that this scientific work offers new original approaches that can be used by scientists in the future. What specific improvements should the authors consider regarding the
methodology? What further controls should be considered?
1. In section 2.1. Materials - purity of all samples should be added. 2. Authors should add used software for calculation pharmacokinetic parameters. 3. Results of HPLC methods are very important in this research. I recommend to present chromatograms with system suitability parameters.    

Author Response

Thank you for your review.

What specific improvements should the authors consider regarding the methodology? What further controls should be considered? 

We wanted to establish a rabbit model with an empty stomach and intestines like fasted rats by focusing on gastric metals, by mild treatment with a low concentration of EDTA 2Na. More than 40 years ago, rabbits with empty stomachs, or stomach-emptying controlled rabbits, were developed. Stomach-emptying controlled rabbits will also have low concentrations of gastric metals. However, it took more than 7 days to develop such rabbits, and the preparation made was a little bit stressful to rabbits. To my knowledge, there is no report focusing on gastric metals and their relationship with drug bioavailability.

  1. In section 2.1. Materials - purity of all samples should be added.

Medicine (LFX, CFX, TC) were used as it is without further purification.

  1. Authors should add used software for calculation pharmacokinetic parameters.

  The calculation was made by using commercially available Excel. This was added in 2.9.5. section.

  1. Results of HPLC methods are very important in this research. I recommend to present chromatograms with system suitability parameters.    

  HPLC charts were added in the text (Figure 3).

Reviewer 5 Report

In current research work, the objective was to perform the oral bioavailability studies of chelating drugs in EDTA-treated rabbits with a lesser amount of gastric metals.

The aim of the study is newer with respect to oral bioavailability studies and experimental work conducted to justify the aim is also exhaustive and in details.

The length of the article is too lengthy, highly recommended to revise the text.

The grammatical language and flow of the article needs to improve for reader.

All the tables and figures of the article needs to thoroughly checked for any mistakes.

Few justifications are required for some results and data obtained from the study to strengthen the objective of the research work, which are highlighted into text.Specific comments are as below:

Title: Needs to revise and be specific with respect to the objective study

Abstract:

1)    Abstract needs to be revise after giving detailed justification for objective, scope of the work, results obtained (require values of Bioavailability) and conclusion drawn from the study.

2)    Line 17: Fasting cannot empty gastric contents because of coprophagic habits. Fasting up to what hrs?  What is the justification for this?  

3)    Line 19: Here, rabbits 19 were kept in fixing boxes to prevent coprophagy and food and ingested >10 mL of 1 mg/mL EDTA 2Na for 1 day before the experiment.—Please revise this sentence structure.

4)    What is the rationale for control rabbits – they were without preventing coprophagy. How? Please justify.

5)    Mention the data for the oral bioavailabilitiies of LFX-EHE and CFX-EHE in EDTA-treated rabbits.

6)    Why Coadministration of Al(OH)3 reduced the oral bioavailabilities of LFX and CFX in and EDTA-treated rabbits? Please justify in brief.

7)    Mention the oral bioavailabilitiies of LFX, CFX, LFX-EHE and CFX-EHE and compare the results. CFX-EHE and LFX-EHE exhibited higher lipophilicity, what is the impact on BA (Bioavability)?

8)    What is outcome of research work? Please mention in detail as food and containing metals affects BA of drugs.

Introduction

1)    Please revise the introduction with targeting specific points like background of research work, problems, objective of research work, scope, methodology targeted, probable results and outcome of the research work.  .

2)    The slow GER can also affects the BA. Is this also a part of research work?

3)    Line 63: In addition, the oral bioavailability of these chelating drugs could be varied among rabbits and foods due to the difference in cationic metal composition in foods among different manufacturing companies. - What measures are taken to control this factor in the current study?

4)    Please mention the information from latest available review articles.  

5)    Please mention the name of probable metals present in rabbit food with reference.

6)    Line 70- In the treatment of rabbits with EDTA 2Na orally, or EDTA-treated rabbits, rabbits were fasted with free access to 1 mg/mL EDTA 2Na solution and prevented coprophagy one day before experiments.- Why EDTA not infused intravenously? What will be the impact if infused intravenously? Please justify.

7)    How coprophagy prevented before one day?

8)    Line 81: CFX-EHE was newly synthesized in the present study. What about LFX-EHE?

9)    Introduction section require to organize with respect to the objective and scope of the study. Currently, flow of the content is not organized.

Materials and methods

1)    The synthetic process of LFX-EHE needs to summarize in Figure.

2)    What is the rationale to study Partition coefficients?

3)    Give reference for Partition coefficients, Solubility study and Chelation with Al3+ ion in vitro.

4)    Partition coefficients, Chelation with Al3+ ion in vitro and solubility study of LFX and LFX-EHE, TC were not performed? Please justify.

5)    Why acidic pH were not used to perform chemical stability?

6)    Please mention the rationale for each methods in brief performed with respect to the objective of the work.

7)    Enzymatic stability of CFX-EHE (the initial concentration was 0.1 mM) was determined at 1, 5, and 15 min after the start of incubation at 37℃ using the following specimen…  Please explain in details.

8)    What is the rationale for wooden rabbit fixing boxes? Explain in brief.

9)    How dose of 1 mg/mL EDTA 2Na fixed?

10)  Please mention the reference for Estimation of Ca and Mg concentrations in gastric contents.

11)  Please verify reference 36 for Evaluation of mucosal damage after EDTA treatment.

12)  How dose of all drug has fixed? Please give reference for that.

13) Line 275: Blood ssamples- Revise this.

14) TC dose for oral and intravenous infusion are different. Please explain.

Results and Discussion

1)    Where is the Partition coefficient and solubility data for LFX?

2)    Please give footnote for 0 in Table 2.

3)    Rewrite this value in table 2- 63.2±21,3

4)    Rewrite this value in table 3- 0. M PBS (pH 7.4)

5)    Table 4 needs revision in terms of presenting data.

6)    Revise table 5 after addition of group description.

7)    Line 486: Revise the title- Oral bioavailability of FD-4 in control and EDTA-treated rabbits

8)    Value of 0,29±0.29 in table 5 – make corrections.

9)    Where is the table for Oral bioavailability of drugs in EDTA-treated rabbits?

10)  Figure 5 and 6. Plasma concentration – time profile … footnotes are not easily understood. (A,B,C and D)

11) How BA of EDTA treated LFX + Al (OH)3 is 119.04±9.37. Please explain.

12) Line 670- developed EDTA-treated rabbits- Revise this.

13)  Discussion part needs to revise with more emphasis on desired results and outcome. Currently, many irrelevant information are presented.

Conclusion

1)    Justify that objective of research work has been satisfied.

2)    How the outcome of the research work will be correlated with clinical study?

Needs improvement 

Author Response

Thank you very much for your kind indication.

Abstract:

1)    Abstract needs to be revised after giving detailed justification for objective, scope of the work, results obtained (require values of Bioavailability) and conclusion drawn from the study.

The revision of the abstract was made.

2)    Line 17: Fasting cannot empty gastric contents because of coprophagic habits. Fasting up to what hrs?  What is the justification for this?  

Usually, it is rreported that 1-2 days of fasting cannot empty rabbits’ stomachs, or “the stomach of a healthy rabbit are never empty”, Sometimes, it is written that rabbits’ stomach is never emptied. [references,1-3]. However, the fasting period is not reported. In my experience, the stomach was still mostly with hair even 3 days-fastin. As discussed in the present study, to establish stomach (or gastric)-emptying controlled rabbits, it took about 7 days [7, 8]. In the present study, the administration of more than 10 mL of 1 mg/mL EFTA 2 Na was found to facilitate the empty of rabbit’s stomach including gastric metals. To my knowledge, there is no report regarding this previously. 

3)    Line 19: Here, rabbits 19 were kept in fixing boxes to prevent coprophagy and food and ingested >10 mL of 1 mg/mL EDTA 2Na for 1 day before the experiment.—Please revise this sentence structure.

Sentences were revised;

Page 8-9: One day before experiments, control rabbits were fasted with free access to water by keeping them in the stainless-steel cages as described above. Coprophagy was not prevented in control rabbits. In the case of EDTA-treated rabbits, rabbits were moved to wooden rabbit fixing boxes (KN-319-A, Natsume Seisakusho  Co., Ltd., Tokyo, Japan) one day before experiments, in which the neck was fixed to prevent food intake and coprophagy but with free access to 1 mg/mL EDTA 2 Na aqueous solution.

Page 21: EDTA-treated rabbits were prepared by keeping rabbits in fixing boxes (in which the neck of rabbits is fixed) to prevent food intake and coprophagy, fasting for one day with free access to 1 mg/mL EDTA 2Na, and ingesting 10 mL EDTA 2Na 1 h before the experiment.

4)    What is the rationale for control rabbits – they were without preventing coprophagy. How? Please justify.

They were kept in the stainless-steal rabbits’ cages as described in the experimental section (Page 8-9). In such free moving cages, rabbits can eat feces from the anus directly.

5)    Mention the data for the oral bioavailabilitiies of LFX-EHE and CFX-EHE in EDTA-treated rabbits.

Regarding the bioavailability of LFX and CFX from LFX-EHE and CFX-EHE, respectively in the presence of Al(OH)3, some comments were added in the test (Page 25 and Page 25-27, respectively).

6)    Why Coadministration of Al(OH)3 reduced the oral bioavailabilities of LFX and CFX in and EDTA-treated rabbits? Please justify in brief.

Both LFX and CFX are representative chelating antibiotics, and they form chelate complexes with Al ions in water. The chelate complexes with Al3+ ions is poorly soluble in water, thus they are not absorbed. In general, only solubilized molecules can be absorbed. The structure of chelate complexes of each medicine is discussed in the discussion section in detail, respectively (6.2, 6.3).

7)    Mention the oral bioavailabilitiies of LFX, CFX, LFX-EHE and CFX-EHE and compare the results.

CFX-EHE and LFX-EHE exhibited higher lipophilicity, what is the impact on BA (Bioavability)?

Page 17: The oral bioavailability of LFX from LFX-EHE was comparable with the oral bioavailability of LFX itself in both control and EDTA-treated rabbits, exhibiting higher oral bioavailabilities.

Page 18: When the oral bioavailability of CFX from CFX-EHE was compared with the oral bioavailability of CFX itself, there was no significant difference between them in both control and EDTA-treated rabbits, as well as the cases of LFX-EHE.

As discussed in the discussion section, both LFX and CFX are thought to be absorbed by influx transporter (OATP)-mediated transport. In contrast, their ester prodrugs are though to be absorbed by passive diffusion, because their lipophilicity is increased. However, there was no much difference in the bioavailability of CFX  (or intestinal absorption) between CFX and CFX-EHE Very strange. Further study is necessary to understand the mechanism.

Regarding this, a short comment was added in the Abstract (Page 2) and discussion section.

Page 2: There was no significant differences in the oral bioavailabilities of LFX and CFX as compared with their EHE prodrugs.

Page 24: There was no difference in the oral bioavailability of LFX between LFX-EHE and LFX itself in both control and EDTA-treated rabbits (Table 8), although the lipophilicity of LFX-EHE was 13-fold higher than LFX [33]. The reason is not clear at present, but both LFX and LFX-EHE are considered to be absorbed efficiently.

Page 26: There was no difference in the oral bioavailability of CFX between CFX-EHE and CFX itself in both control and EDTA-treated rabbits, irrespective of 13-fold higher lipophilicity (PC) of CFX-EHE than CFX itself (Tables 1 and  9). In this case, CFX-EHE is considered to be absorbed by passive diffusion, and CFX itself is considered to be absorbed mainly by OATP-mediated transport as discussed already. Further study is necessary to clarify the mechanism, why there is no difference in the oral bioavailability of CFX between CFX and CFX-EHE?

8)    What is outcome of research work? Please mention in detail as food and containing metals affects BA of drugs.

The outcome of present research work is thought to be the development of rabbits model with almost empty gastric contents, gastric metals and low gastric pH, as well as the cases of fasted rats and fasted humans, by very simple method (administration of low concentration EDTA 2Na) without causing mucosal damage.

A short comment was added in the abstract and conclusion as follows;

In conclusion, EDTA-treated rabbits can avoid the effect of chelating in the stomach in preclinical oral bioavailability studies of chelating drugs as well as the case in fasted rats, because the gastric metals of rabbits are eliminated,

Introduction

1)    Please revise the introduction with targeting specific points like background of research work, problems, objective of research work, scope, methodology targeted, probable results and outcome of the research work.  .

Introduction section was revised by consideringyour suggestion.

2)    The slow GER can also affects the BA. Is this also a part of research work?]

In this study, the effect of gastric metals on oral bioavailability of chelating drugs was mainly considered.

However, by treating rabbits with EDTA 2Na, the gastric contents were found to be eliminated greatly, suggesting that the problem of slow GER in oral bioavailability study in rabbits will also be mostly eliminated in EDTA-treated rabbits. However, in the present study, we didn’t evaluate the effect of GER on oral bioavailability of drugs in detail. In the case of water soluble drugs such as LFX and CFX, the slow GER ddoesn’t affect much on their oral bioavailability.

3)    Line 63: In addition, the oral bioavailability of these chelating drugs could be varied among rabbits and foods due to the difference in cationic metal composition in foods among different manufacturing companies. - What measures are taken to control this factor in the current study?

As discussed in the discussion section, the composition of cationic metals in rabbit’s food was found to be fairly different among companies (we got information on composition from the manufacturing company, directly), although we didn’t write the composition in detail in the present study. It will be difficult to evaluate the food composition correctly, because each metal has biological role for rabbit health. EDTA-treated rabbits can eliminate such the effect of food cationic metals on oral bioavailability temporarily as well as the case in fasted rats, fasted dogs and humans.

4)    Please mention the information from latest available review articles.  

To my knowledge, there are no articles or reviews describing the effect of gastric metals on drug bioavailability in rabbits. I think (hope) that our present article is the first one.

5)    Please mention the name of probable metals present in rabbit food with reference.

The composition of metals in rabbit foods was described in the discussion section (6.1). However, we didn’t describe all data that we did know from the company privately. The name of commercial products and company name is listed in the Discussion section. The composition of Oriental Yeast Co. is open on the Internet (Japanese only).

6)    Line 70- In the treatment of rabbits with EDTA 2Na orally, or EDTA-treated rabbits, rabbits were fasted with free access to 1 mg/mL EDTA 2Na solution and prevented coprophagy one day before experiments.- Why EDTA not infused intravenously? What will be the impact if infused intravenously? Please justify.

In our study, EDTA was administered orally to eliminate gastric metals by chelate formation. In EDTA chelation therapy, EDTA is infused intravenously to eliminate metal cations in the body (blood and tissues) not of gastrointestinal tracts. EDTA 2Na is a hydrophilic compound. Thus, the transport of EDTA itself from the intestine to blood or from blood to the intestine will be small, if any.

7)    How coprophagy prevented before one day?

Rabbits were kept in a commercially available fixing boxes, in which the neck of rabbits is fixed, not free moving.

8)    Line 81: CFX-EHE was newly synthesized in the present study. What about LFX-EHE?

CFX-EHE was newly synthetized in the present study. LFX-EHE and some other ester prodrugs of LFX were synthetized previously, and their physicochemical properties and some oral bioavailability studies were reported [Ref 33]. However, oral bioavailability studies in control and EDTA-treated rabbits as described in the present study were performed newly in the present study (the treatment method of rabbits is different).

9)    Introduction section require to organize with respect to the objective and scope of the study. Currently, flow of the content is not organized.

Correction of introduction was made according to the suggestion (1).

Materials and methods

1)    The synthetic process of LFX-EHE needs to summarize in Figure.

LFX-EHE was synthetized in the same manner as reported previously in our published article [Ref 33]. LFX-EHE is not a new compound for us, different from CFX-EHE. CFX-EHE will not be reported so far.

2)    What is the rationale to study Partition coefficients?

To measure lipophilicity. Lipophilic compounds are expected to be absorbed by passive diffusion from anywhere in the small intestine. In contrast, most of hydriophilic compounds such as amino acids, glucose and also quinolone antibiotics such as LFX and CFX are absorbed by specific influx transporter-mediated transport. The expression site of such transporter is site dependent. Depending on the lipophilicity, the absorption mechanism can be altered, although, in the present study, the transport mechanism of ester prodrugs was not examined in detail. (It maybe interesting to examine the substrate specificity of ester prodrugs for efflux transporter P-gp)

3)    Give reference for Partition coefficients, Solubility study and Chelation with Al3+ ion in vitro.

The study was made in the same manner as reported previously [Ref 32, 33], References were added in the text,

4)    Partition coefficients, Chelation with Al3+ ion in vitro and solubility study of LFX and LFX-EHE, TC were not performed? Please justify.

Regarding the PC and chelation of LFX and LFX-EHE were already reported in our previous report [Ref 33]. Thus, their values are described in the text of each section (3.2.1 – 3.2.4).

5)    Why acidic pH were not used to perform chemical stability?

Ester prodrugs are stable in acidic conditions.

6)    Please mention the rationale for each method in brief performed with respect to the objective of the work.

A short comment was added to each method (3.2.1. – 3.2.4).

7)    Enzymatic stability of CFX-EHE (the initial concentration was 0.1 mM) was determined at 1, 5, and 15 min after the start of incubation at 37℃ using the following specimen…  Please explain in details.

A short comment was added: indicating that CFX-EHE can produce an active parent drug, CFX, rapidly when CFX-EHE was aabsorbed into the intestinal membrane. Prodrugs of antibiotics should be converted to their pharmacologically active parent compounds rapidly. 

8)    What is the rationale for wooden rabbit fixing boxes? Explain in brief.

In rabbit fixing boxes, the neck of the rabbit was fixed. Thus, coprophagy was prevented, because rabbits eat feces from the anus directly This was added in page 8-9 in red.

9)    How dose of 1 mg/mL EDTA 2Na fixed?

Regarding the dose (concentration) setting of EDTA 2Na, a long discussion was made in Discussion section (6. 1). Lower but effective chelating activity of EDTA 2Na was necessary to remove gastric metals.

10)  Please mention the reference for Estimation of Ca and Mg concentrations in gastric contents.

The concentrations of Ca and Mg in plasma are generally determined by using assay kit in clinical. In the present study, we used such clinically available assay kit to estimate Ca and Mg concentrations in gastric contents. In the present study, gastric metals concentrations in rabbits were determined by using such assay kit. There is no specific reference for this, because there is no report regarding the concentrations of gastric metals so far. 

11)  Please verify reference 36 for Evaluation of mucosal damage after EDTA treatment.

In general, chelating compounds can expand the paracellular route, and therefore membrane permeability to  macromolecule compounds such as FD-4 is increased. However, the intestinal absorption of FD-4 didn’t change by EDTA treatment. These are discussed in Discussion section (6.1). For FD-4, a new reference was added [36].

12)  How dose of all drug has fixed? Please give reference for that.

There is no reference. But the dose for humans is listed in Table 11. To increase the sensitivity of analysis, a higher dose (approximately twice of human) was used. In the case TC (more than 30-fold of human), to set the dose, preliminary study was made.

13) Line 275: Blood ssamples- Revise this.

It was revised.

14) TC dose for oral and intravenous infusion are different. Please explain.

In the case of intravenous infusion, bioavailability is always 100% of dose (100% of dosed amount enters into the entral blood circulation). But in the case of oral administration of TC in rabbits, the bioavailability was less than 5%. The low oral bioavailability in rabbits was thought to be due to the adsorptive interaction, although further detailed study is necessary. These are commented in Discussion section.

Results and Discussion

1)    Where is the Partition coefficient and solubility data for LFX?

    The data are described in the text as sentences (3.2.1), because data were already reported in our previous report [32, 33].

2)    Please give footnote for 0 in Table 2.

   A short comment was added in Table 2.

3)    Rewrite this value in table 2- 63.2±21,3

   Correction was made in Table 2.

4)    Rewrite this value in table 3- 0. M PBS (pH 7.4)

  It was 0.1M. Correction was made.

5)    Table 4 needs revision in terms of presenting data.

The title of Table 4 was revised.

6)    Revise table 5 after addition of group description.

Table 5 was revised. Title was also altered.

7)    Line 486: Revise the title- Oral bioavailability of FD-4 in control and EDTA-treated rabbits

The title was revised.

8)    Value of 0,29±0.29 in table 5 – make corrections.

Correction was made.

9)    Where is the table for Oral bioavailability of drugs in EDTA-treated rabbits?

Oral bioavailabilities of drugs in EDTA-treated rabbits are listed in Tables 8-10.

10)  Figure 5 and 6. Plasma concentration – time profile … footnotes are not easily understood. (A,B,C and D)

Figs. 5 and 6 and the footnotes were revised.

11) How BA of EDTA treated LFX + Al (OH)3 is 119.04±9.37. Please explain.

  To estimate AUC after the final sampling time, AUCt -, the elimination slope (β) in the terminal phase is used for calculation as follows; AUCt - = Ct/β. The small changes (or a little bit low value of β) of β can produce big errors, especially for drugs with high oral bioavailability such as LFX (this drug is considered to be almost 100% bioavailability in humans). This will be not very good, but not the rare case, For example, the absolute oral bioavailability of LFX in sheep reported was 114 ± 27.7% [63]. Similar to our case.

.

12) Line 670- developed EDTA-treated rabbits- Revise this.

Corresponding sentences were revised.

13)  Discussion part needs to revise with more emphasis on desired results and outcome. Currently, many irrelevant information are presented.

   Discussion section was revised.

Conclusion

1)    Justify that objective of research work has been satisfied.

     Positive data were described.

2)    How the outcome of the research work will be correlated with clinical study?

   EDTA-treated rabbits is expected to be available in preclinical study, although still marked inter-species difference was observed in some potent adsorptive drugs such as TC and CFX.

   On one hand, EDTA-treated rabbits with small gastric contents, gastric metals and low gastric pH will be useful animal models for many of other drugs with low adsorptive potency such as LFX.